# Coupling a three-dimensional subsurface flow and transport model with a land surface model to simulate stream-aquifer-land interactions (CP v1.0)

Gautam Bisht[1], Maoyi Huang[2,*], Tian Zhou[2], Xingyuan Chen[2], Heng Dai[2], Glenn Hammond[3], William Riley[1], Janelle Downs[2], Ying Liu[2], John Zachara[2]

[1]Lawrence Berkeley National Laboratory, Berkeley, CA

[2]Pacific Northwest National Laboratory, Richland, WA

[3]Sandia National Laboratories, Albuquerque, NM

Correspondence to: Maoyi Huang (maoyi.huang@pnnl.gov)

Revised Manuscript to be considered for *Geoscientific Model Development*

**Abstract**

A fully coupled three-dimensional surface and subsurface land model is developed and applied to a site along the Columbia River to simulate three-way interactions among river water, groundwater, and land surface processes. The model features the coupling of the Community Land Model version 4.5 (CLM4.5) and a massively-parallel multi-physics reactive transport model (PFLOTRAN). The coupled model, named CP v1.0, is applied to a 400 m × 400 m study domain instrumented with groundwater monitoring wells along the Columbia River shoreline. CP v1.0 simulations are performed at three spatial resolutions (i.e., 2 m, 10 m, and 20 m) over a five-year period to evaluate the impact of hydro-climatic conditions and spatial resolution on simulated variables. Results show that the coupled model is capable of simulating groundwater-river water interactions driven by river stage variability along managed river reaches, which are of global significance as a result of over 30,000 dams constructed worldwide during the past half century. Our numerical experiments suggest that the land-surface energy partitioning is strongly modulated by groundwater-river water interactions through expanding the periodically inundated fraction of the riparian zone, and enhancing moisture availability in the vadose zone via capillary rise in response to the river stage change. Furthermore, spatial resolution is found to impact significantly the accuracy of estimated the mass exchange rates at the boundaries of the aquifer, and it becomes critical when surface and subsurface become more tightly coupled with groundwater table within six to seven meters below the surface. Inclusion of lateral subsurface flow influenced both the surface energy budget and subsurface transport processes as a result of river water intrusion into the subsurface in response to elevated river stage that increased soil moisture for evapotranspiration and suppressed available energy for sensible heat in the warm season. The coupled model developed in this study can be used for improving mechanistic understanding of ecosystem functioning and biogeochemical cycling along river corridors under historical and future hydro-climatic changes. The dataset presented in this study can also serve as a good benchmarking case for testing other integrated models.

## 1  Introduction

Previous modeling studies have demonstrated that subsurface hydrologic model structure and parameterization can significantly affect simulated land-atmosphere exchanges [*Condon et al.*, 2013; *Hou et al.*, 2012; *Kollet and Maxwell*, 2008; *Miguez-Macho and Fan*, 2012] and therefore boundary layer dynamics [*Maxwell and Miller*, 2005; *Rihani et al.*, 2015], cloud formation [*Rahman et al.*, 2015], and climate [*Leung et al.*, 2011; *Taylor et al.*, 2013]. Lateral subsurface processes are fundamentally important at multiple spatial scales, including hill-slope scales [*McNamara et al.*, 2005; *Zhang et al.*, 2011], basin scales in semi-arid and arid climates where regional aquifers sustain baseflows in rivers [*Schaller and Fan*, 2009], and wetlands [*Fan and Miguez-Macho*, 2011]. However, some current-generation land surface models (LSMs) routinely omit explicit lateral subsurface processes [*Clark et al.*, 2015; *Kollet and Maxwell*, 2008; *Nir et al.*, 2014], while others include them (described below). Observational and modeling studies suggest that groundwater forms an environmental gradient in soil moisture availability by redistributing water that could profoundly shape critical zone evolution at continental to global scales [*Fan et al.*, 2013; *Taylor et al.*, 2013]. The mismatch between observed and simulated evapotranspiration by current LSMs could be explained by the absence of lateral groundwater flow [*Maxwell and Condon*, 2016].

It has been increasingly recognized that rivers, despite their small aerial extent on the landscape, play important roles in watershed functioning through their connections with groundwater aquifers and riparian zones [*Shen et al.*, 2016]. The interactions between groundwater and river water prolong physical storage and enhance reactive processing that alter water chemistry, downstream transport of materials and energy, and biogenic gas emissions [*Fischer et al.*, 2005; *Harvey and Gooseff*, 2015]. The Earth System modeling community recognizes such a gap in existing Earth system models and calls for improved representation of biophysical and biogeochemical processes within the terrestrial-aquatic interface [*Gaillardet et al.*, 2014].

Over the past decade, much effort has been expended to include groundwater into LSMs. Groundwater is important to water and energy budgets such as evapotranspiration (ET), latent heat (LH), and sensible heat (SH), but also to biogeochemical processes such as gross primary production, heterotrophic respiration, and nutrient cycling. The lateral convergence of water along the landscape and two-way groundwater-surface water exchange are identified as the most

relevant subsurface processes to large-scale Earth System functioning (see review by *Clark et al.*
[2015]). However, the choice of processes, the approaches to represent multi-scale structures and
heterogeneities, the data and computational demands, etc., all vary greatly among the research
groups even working on the same land models.
Most of the LSMs reviewed by *Clark et al.* [2015] do not explicitly account fort stream-
aquifer-land interactions. For example, the Community Land Model version 4.5 allows for
reinfiltration of flooded waters in a highly parameterized way without explicitly linking to
groundwater dynamics, therefore only one-way flow from the aquifer to the stream is simulated
[*Oleson et al.*, 2013]. The Land-Ecosystem-Atmosphere Feedback model treats river elevation as
part of the 2-D vertically integrated groundwater flow equation and allows river and floodwater
to infiltrate through sediments in the flood plain [*Miguez-Macho and Fan*, 2012].
In contrast, the fully integrated models, being a small subset of LSMs, explicitly represent
the two-way exchange between groundwater aquifers and their adjacent rivers in a spatially
resolved fashion. Such models couple a completely integrated hydrology model with a land
surface model, so that the surface-water recharge to groundwater by infiltration or intrusion and
base flow discharge from groundwater to surface waters can be estimated in a more mechanistic
way.
Examples of the integrated models include: (1) the coupling between the Common Land
Model (CoLM) and a variably saturated groundwater model (ParFlow) [*Maxwell and Miller*,
2005]; (2) the Penn State Integrated Hydrologic Model (PIHM) [*Shi et al.*, 2013]; (3) the
coupling between the Process-based Adaptive Watershed Simulator (PAWS) and CLM4.5 [*Ji et*
*al.*, 2015; *Pau et al.*, 2016; *Riley and Shen*, 2014]; (4) the coupling between the CATchment
HYdrology (CATHY) model and the Noah model with multiple parameterization schemes
(Noah-MP) [*Niu et al.*, 2014]; and (5) the coupling between CLM3.5 and ParFlow through the
Ocean Atmosphere Sea Ice Soil external coupler (OASIS3) in the Terrestrial Systems Modeling
Platform (TerrSysMP) [*Shrestha et al., 2014*; *Gebler et al., 2017*]. The integrated models
eliminate the need for parameterizing lateral groundwater flow and allow the interconnected
groundwater–surface-water systems to evolve dynamically based on the governing equations and
the properties of the physical system. Although such models often require robust numerical
solvers on high-performance computing (HPC) facilities to achieve high-resolution, large-extent
simulations [*Maxwell et al.*, 2015], they have been increasingly applied for hydrologic prediction
and environmental understanding. However, as a result of difference in physical process
representations and numerical solution approaches in terms of (1) the coupling between the
variably saturated groundwater and surface water flow; (2) representation of surface water flow;
and (3) implementation of subsurface heterogeneity in the existing integrated models,
significant discrepancies exist in their results when the models were applied to highly nonlinear
problems with heterogeneity and complex water table dynamics, while many of the models show
good agreement for simpler test cases where traditional runoff generation mechanisms (i.e.,
saturation and infiltration excess runoff) apply [*Kollet et al.*, 2017; *Maxwell et al.*, 2014].
The developments of the integrated models have enabled scientific explorations of
interactions and feedback mechanisms in the aquifer-soil-vegetation-atmosphere continuum
using a holistic and physically based approach [*Shrestha et al.*, 2014; *Gilbert et al.*, 2017].
Compared to simulations of regional climate models coupled to traditional LSMs, such a
physically based approach shows less sensitivity to uncertainty in the subsurface hydraulic
characteristics that could propagate from deep subsurface to free troposphere [*Keune et al.*,
2016], while other physical representations (e.g., parameterizations in evaporation and
transpiration, atmospheric boundary layer schemes) could have significant effects on the
simulations as well [*Sulis et al.*, 2017]. Therefore, it is of great scientific interest to further
develop the integrated models and benchmarks to achieve improved understanding of complex
interactions in the fully coupled Earth system.
Motivated by the great potentials of using an integrated model to explore Earth system
dynamics, the objective of this study is three-fold. First, we aim to document the development of
a coupled land surface and subsurface model as a first step toward a new integrated model,
featuring the two-way coupling between two highly-scalable and state-of-the-art open-source
codes: CLM4.5 [*Oleson et al.*, 2013] and a reactive transport model PFLOTRAN [*Lichtner et al.*,
2015]. The coupled model mechanistically represents the two-way exchange of water and solute
mass between aquifers and river, as well as land-atmosphere exchange of water and energy. The
coupled model is therefore named as CP v1.0 hereafter. We note that in recent years, efforts have
been made to implement carbon–nitrogen decomposition, nitrification, denitrification, and plant
uptake from CLM4.5 in the form of a reaction network solved by PFLOTRAN to enable the
coupling of biogeochemical processes between the two models [*Tang et al.*, 2016]. In addition,
although PAWS is coupled to the same version of CLM (i.e., CLM4.5) [*Ji et al.*, 2015; *Pau et*
*al.*, 2016], PFLOTRAN resolves the subsurface in a 3-D fashion, while PAWS approximates the
3D Richards equation by divide the subsurface into an unsaturated domain represented by the 1-
D Richards Equation coupled with 3D saturated groundwater flow equation for subsurface flow,
by assuming that there is no horizontal flow in unsaturated portion of soil, and that lateral flux in
saturated portion is evenly distributed.
Second, we describe a numerically challenging benchmarking case for verifying coupled land
surface and subsurface models, featuring a highly dynamic river boundary condition determined
by dam-induced river stage variations (Hauer et al., 2017), representative of managed river
reaches that are of global significance as a result of dam constructions in the past few decades
[*Zhou et al.*, 2016]. Third, we assess the effects of spatial resolution and projected hydro-climatic
changes on simulated land surface fluxes and exchange of groundwater and river water using the
coupled model and datasets from the benchmarking case.  In section 2, we describe the
component models and our coupling strategy. In section 3, we describe an application of the
model to a field site along the Hanford reach of the Columbia River, where the subsurface
properties are well characterized and long-term monitoring of river stage, groundwater table, and
exchange of groundwater and river water exist. In section 4, we assess the effects of spatial
resolution and hydro-climatic conditions to simulated fluxes and state variables. In section 5,
conclusion and future work are discussed.

**2    Model description**
**2.1    The Community Land Model version 4.5**
CLM4.5 [*Oleson et al.*, 2013] is the land component of the Community Earth System Model
version 1 (CESM1) [*Hurrell et al.*, 2013], a fully coupled numerical simulator of the Earth
system consisting of atmospheric, ocean, ice, land surface, carbon cycle, and other components.
It has been applied successfully to explore interactions among water, energy, carbon, and
biogeochemical cycling at local to global scales [*Leng et al.*, 2016b; *Xu et al.*, 2016], and proven
to be highly scalable on leading HPC facilities such as the U.S. Department of Energy
(USDOE)'s National Energy Research Scientific Computing Center (NERSC).  The model
includes parameterizations of terrestrial hydrological processes including interception,
throughfall, canopy drip, snow accumulation and melt, water transfer between snow layers,
infiltration, evaporation, surface runoff, sub-surface drainage, redistribution within the soil
column, and groundwater discharge and recharge to simulate changes in canopy water, surface
water, snow water, soil water, and soil ice, and water in the unconfined aquifer [*Oleson et al.*,
2013]. Precipitation is either intercepted by the canopy, falls directly to the snow/soil surface
(throughfall), or drips off the vegetation (canopy drip). Water input at the land surface, the sum
of liquid precipitation reaching the ground and melt water from snow, is partitioned into surface
runoff, surface water storage, and infiltration into the soil. Two sets of runoff generation
parameterizations, including formulations for saturation and infiltration excess runoff and
baseflow, are implemented into the model: the TOPMODEL-based runoff generation
formulations [*Beven and Kirkby*, 1979; *Niu et al.*, 2005; *Niu et al.*, 2007] and the Variable
Infiltration Capacity (VIC)-based runoff generation formulations [*Lei et al.*, 2014; *Liang et al.*,
1994; *Wood et al.*, 1992]. Surface water storage and outflow in and from wetlands and small sub-
grid scale water bodies are parameterized as functions of fine-spatial-scale elevation variations
called microtopography. Soil water is predicted from a multi-layer model based on the 1-D
Richards equation, with boundary conditions and source/sink terms specified as infiltration,
surface and sub-surface runoff, gradient diffusion, gravity, canopy transpiration through root
extraction, and interactions with groundwater. A groundwater component is added in the form of
an unconfined aquifer lying below the soil column following *Niu et al.* [2007]. The model
computes surface energy fluxes following the Monin-Obukhov Similarity Theory using
formulations in Zeng et al. (1998), which updates the calculation of boundary resistance to
account for understory turbulence, sparse and dense canopies, and surface litter layer (Sakaguchi
and Zeng, 2009;Zeng et al., 2005;Zeng and Wang, 2007). Water and energy budgets are
conserved at every modeling step.

## 2.2  PFLOTRAN


PFLOTRAN is a massively-parallel multi-physics simulator [*Hammond et al.*, 2014] developed
and distributed under an open source GNU LGPL license and is freely available through
Bitbucket ((https://bitbucket.org/pflotran/pflotran)). It solves a system of generally nonlinear
partial differential equations (PDEs) describing multiphase, multicomponent and multiscale
reactive flow and transport in porous materials. The PDEs are spatially discretized using a finite
volume technique, and the backward Euler scheme is used for implicit time discretization. It has
been widely used for simulating subsurface multiphase flow and reactive biogeochemical
transport processes [*Chen et al.*, 2013; *Chen et al.*, 2012; *Hammond and Lichtner*, 2010;
*Hammond et al.*, 2011; *Kumar et al.*, 2016; *Lichtner and Hammond*, 2012; *Liu et al.*, 2016; *Pau*
*et al.*, 2014]
PFLOTRAN is written in object-oriented Fortran 2003/2008 and relies on the PETSc
framework [*Balay et al.*, 2015] to provide the underlying parallel data structures and solvers for
scalable high performance computing.   PFLOTRAN uses domain decomposition and MPI
libraries for parallelization. PFLOTRAN has been run on problems composed of over 3 billion
degrees of freedom with up to 262,144 processors, but it is more commonly employed on
problems with millions to tens of millions of degrees of freedom utilizing hundreds to thousands
of processors. Although PFLOTRAN is designed for massively parallel computation, the same
code base can be run on a single processor without recompiling, which may limit problem size
based on available memory.
In this study, PFLOTRAN is used to simulate single phase variably saturated flow and solute
transport in the subsurface.   Single-phase variably saturated flow is based on the Richards
equation with the form
$$\frac{\partial}{\partial t}(\varphi s \rho) + \nabla \cdot \rho \boldsymbol{q} = 0, \tag{1}$$

with liquid density $\rho$, porosity $\varphi$, and saturation $s$.  The Darcy velocity, $\boldsymbol{q}$ , is given by
$$\boldsymbol{q} = -\frac{k k_r}{\mu}\nabla(p - \rho g \boldsymbol{z}), \tag{2}$$

with liquid pressure $p$, viscosity $\mu$, acceleration of gravity $g$, intrinsic permeability $k$, relative
permeability $k_r$ and elevation above a given datum $z$.  Conservative solute transport in the liquid
phase is based on the advection-dispersion equation
$$\frac{\partial}{\partial t}(\varphi s C) + \nabla \cdot (\boldsymbol{q} - \varphi s D \nabla)C = 0, \tag{3}$$

with solute concentration $C$ and hydrodynamic dispersion coefficient $D$.  The discrete system of
nonlinear PDEs for flow and transport are solved using the Newton-Raphson method.

## 2.3 Model coupling

In this study, CLM4.5's one-dimensional models for flow in unsaturated [*Zeng and Decker*, 2009] and saturated [*Niu et al*., 2007] zones are replaced by PFLOTRAN's RICHRADS mode to simulate unsaturated-saturated flow within the three-dimensional subsurface domain. Although PFLOTRAN is also capable of simulating coupled flow and thermal processes in the subsurface including explicit representation of liquid-ice phase [*Karra et al*., 2014], as well as, soil nutrient cycles [*Hammond and Lichtner*, 2010; *Zachara et al*., 2016; *Tang et al*., 2016], those processes are not coupled between the two models in this study. A schematic representation of the coupling between CLM4.5 and PFLOTRAN is shown in Figure 1. A model coupling interface based on PETSc data structures was developed to couple the two models and the interface includes some key design features of the CESM coupler [*Craig et al.*, 2012]. The model coupling interface allows each model grid to have a different spatial resolution and domain decomposition across multiple processors. While CLM4.5 uses a round-robin decomposition approach, PFLOTRAN employs domain decomposition via PETSc (Figure 1a). Interpolation of gridded data from one model onto the grids of the other is done through sparse matrix vector multiplication. As a preprocessing step, sparse weight matrices for interpolating data between the two models are saved as mapping files. Analogous to the CESM coupler, the mapping files are saved in a format similar to the mapping files produced by the ESMF_RegridWeightGen (https://www.earthsystemcog.org/projects/regridweightgen). ESMF regridding tools provide multiple interpolation methods (conservative, bilinear, and nearest neighbor) to generate the sparse weight matrix.

In this work, we have used a conservative remapping method to interpolate data between CLM and PFLOTRAN. During model initialization, the model coupling interface first collectively reads all required sparse matrices. Next, the model coupling interface reassembles local sparse matrices after accounting for domain decomposition of each model (figures 1b and 1c). . For a given time step, CLM4.5 first computes infiltration, evaporation, and transpiration within the domain and then sends the data to the model coupling interface. The model coupling interface for each processor receives relevant CLM data vector from all other processors; interpolates data from CLM's grid onto PFLOTRAN's grid via a local sparse matrix vector multiplication; and saves the resulting vector in PFLOTRAN's data structures as prescribed flow conditions (Figure 1b). PFLOTRAN evolves the subsurface states over the given time step

length. The updated soil moisture simulated by PFLOTRAN are then provided back to the model
coupling interface, which interpolates data from PFLOTRAN's grid onto CLM's grid (Figure
1c). The interpolated data is saved in CLM4.5's data structure and used for simulating land
water- and energy- budget terms in the next step. Figure 2 shows a schematic representation of
how stream-aquifer-land interactions are simulated in CP v1.0 when applied to the field scale,
such as the 300 Area domain to be introduced in section 3.1.

## 261    3    Site description and model configuration

### 262    3.1   The Hanford site and the 300 Area

The Hanford Reach is a stretch of the lower Columbia River extending approximately 55 km
from the Priest Rapids hydroelectric dam to the outskirts of Richland, Washington, USA (Figure
3a) [*Tiffan et al.*, 2002]. The Columbia River above Priest Rapids Dam drains primarily
mountainous regions in Canada, Idaho, Montana, and Washington, over which spatio-temporal
distributions of precipitation and snowmelt modulate the timing and magnitude of river flows
[*Elsner et al.*, 2010; *Hamlet and Lettenmaier*, 1999]. The Columbia River is highly regularted by
dams for power generation and river stage and discharge along the Hanford Reach displays
significant variation on multiple time scales. Strong seasonal variations occur with the greatest
discharge (up to 12,000 $m^3$ $s^{-1}$) occurring from May through July due to snow melt, with less
discharge (>1,700 $m^3$ $s^{-1}$) and lower flows occurring in the fall and winter [*Hamlet and*
*Lettenmaier*, 1999; *Waichler et al.*, 2005]. Significant variation in discharge also occurs on a
daily or hourly basis due to power generation, with fluctuations in river stage of up to 2 m within
a 6-24 hr period being common [*Tiffan et al.*, 2002].

276        The Hanford site features an unconfined aquifer developed in Miocene-Pliocene fluvial and

lacustrine sediments of the Ringold Formation, overlain by Pleistocene flood gravels of the
Hanford formation [*Thorne et al.*, 2006] that is in hydrologic continuity with the Columbia
River. The Hanford formation gravel and sand, deposited by glacial outburst floods at the end of
the Pleistocene [*Bjornstad*, 2007], has a high average hydraulic conductivity at ~3,100 m day$^{-1}$
[*Williams et al.*, 2008]. The fluvial deposits of the Ringold Formation have much lower
hydraulic conductivity than the Hanford but are still relatively conductive at 36 m day$^{-1}$
[*Williams et al.*, 2008]. Fine-grained lacustrine Ringold silt has a much lower estimated
hydraulic conductivity of 1 m day$^{-1}$. The hydraulic conductivity of recent alluvium lining the
river channel is low relative to the Hanford formation, which tends to dampen the response of
water table elevation in wells near the river when changes occur in river stage [*Hammond et al.*,
2011; *Williams et al.*, 2008]. Overall, the Columbia River through the Hanford Reach is a prime
example of a hyporheic corridor with an extensive floodplain aquifer. It is consequently an ideal
alluvial system for evaluating the capability of the coupled model in simulating stream-aquifer-
land interactions.
The region is situated in a cold desert climate with temperatures, precipitation, and winds that
are greatly affected by the presence of mountain barriers. The Cascade Range to the west creates
a strong rain shadow effect by forming a barrier to moist air moving from the Pacific Ocean,
while the Rocky Mountains and ranges to the north protect it from the more severe cold polar air
masses and winter storms moving south across Canada. Meteorological data are collected by the
Hanford Meteorological Monitoring Network (http://www.hanford.gov/page.cfm/hms), which
collects meteorological data representative of the general climatic conditions for the Hanford
site.
A segment of the hyporheic corridor in the Hanford 300 Area (300A) was chosen to evaluate
the model's capability in simulating river-aquifer-land interactions. Located at the downstream
end of the Hanford Reach, the impact of dam operations on river stage is relatively damped,
exhibiting a typical variation of ~0.5 m within a day and 2-3 m in a year. The study domain
covers an area of 400 m × 400 m along the Columbia River shoreline (Figure 3b). Aquifer
sediments in the 300 Area are coarse grained and highly permeable [*Chen et al., 2013*;
*Hammond and Lichtner, 2010*]. Coupled with dynamic river stage variations, the resulting
system is characterized by stage-driven intrusion and retreat of river water into the adjacent
unconfined aquifer system. During high-stage spring runoff events, river water has been detected
in monitoring wells nearly 400 m from the shoreline [*Williams et al.*, 2008]. During baseline,
low-stage conditions (October-February), the Columbia River is a gaining stream, and the
aquifer pore space is occupied by groundwater.
The study domain is instrumented with groundwater monitoring wells (Figure 3b) and a river
gaging station that records water table elevations. A vegetation survey in 2015 was conducted to
provide aerial coverages of grassland, shrubland, riparian trees in the domain (Figure 3b). A
high-resolution topography and bathymetry dataset at 1-m resolution was assembled from
multiple surveys by *Coleman et al.* [2010]. The data layers originated from Deep Water
Bathymetric Boat surveys, terrestrial Light Detection and Ranging (LiDAR) surveys, and special
hydrographic LiDAR surveys penetrating through water to collect both topographic and
bathymetric elevation data.

## 3.2   Model configuration, numerical experiments, and analyses

To assess the effect of spatial resolution on simulated variables such as latent heat, sensible heat,
water table depth, and river water in the domain, we configured CP v1.0 simulations at three
horizontal spatial resolutions: 2-m, 10-m, and 20-m over the 400 m×400 m domain, respectively.
For comparison purposes, we also configured a 2-m-resolution CP v1.0 vertical only simulation
(i.e., $S_{v2m}$) in which lateral transfers of flow and solutes in the subsurface are disabled. Due to
lack of observations of water and energy fluxes from the land surface, in this study we treat the
2-m-resolution CP v1.0 as the baseline and compare simulation results at other resolutions to it.
New hydrologic regimes are projected to emerge over the Pacific Northwest in as early as the
2030s due to increases in winter precipitation and earlier snow melt in response to future
warming [*Leng et al.*, 2016a]. Therefore, we expect that spring and early summer river discharge
along the reach might increase in the future. To evaluate how land surface-subsurface coupling
might be modulated hydro-climatic conditions, we designed additional numerical experiments by
driving the model with elevated river stages by adding five meters to the observed river stage
time series. The simulations and their configurations are summarized in Table 1.

334       The PFLOTRAN subsurface domain, also terrain-following and extending from soil surface
(including riverbed) to 32 m below the surface, was discretized using a structured approach with
rectangular grids. For the 2-m, 10-m, and 20-m resolution simulations, each mesh element was 2
m × 2 m, 10 m × 10 m, and 20 m×20 m, in the horizontal direction, and 0.5 m in the vertical
direction, giving $2.56 \times 10^6$, $99.2 \times 10^3$, and $2.48 \times 10^3$ control volumes in total. The domain
contained two materials with contrasting hydraulic conductivities: Hanford and Ringold (Figure
4). Note that only the soil moisture and soil hydraulic properties within the top 3.8 m are
transferred from PFLOTRAN to CLM4.5 to allow simulations of infiltration, evaporation, and
transpiration in the next time step, as the CLM4.5 subsurface domain is limited to 3.8 meters and
cannot currently be easily modified. The hydrogeological properties of the Hanford and Ringold
materials (Table 2) were taken from *Williams et al.* [2008]. The unsaturated hydraulic
conductivity in PFLTORAN simulations was computed using the Van Genuchten water retention
function [*van Genuchten*, 1980] and the Burdine permeability relationship [*Burdine*, 1953].
We applied time varying pressure boundary conditions to PFLOTRAN's subsurface domain
at the northern, western, and southern boundaries. The transient boundary conditions were
derived using kriging-based interpolations of hourly water table elevation measurements in wells
inside and beyond the model domain, following the approach used by *Chen et al.* [2013].
Transient head boundary conditions were applied at the eastern boundary with water table
elevations from the river gaging station and the gradient along the river estimated using water
elevations simulated by a 1-D hydraulic model along the reach, the Modular Aquatic Simulation
System in 1-Dimension (MASS1) [*Waichler et al.*, 2005], with a Nash–Sutcliffe coefficient
[*Nash and Sutcliffe*, 1970] of 0.99 in the simulation period (figure not shown). The river stage
simulated by MASS1 was also used to fill river stage measurement gaps caused by instrument
failures. A conductance value of $10^{-12}$ m was applied to the eastern shoreline boundary to mimic
the damping effect of low-permeability material on the river bed [*Hammond and Lichtner*,
2010]. A no-flow boundary condition was specified at the bottom of the domain to represent the
basalt underlying the Ringold formation.
Vegetation types (Figure 3b) were converted to corresponding CLM4.5 plant functional types
(PFTs) and bare soil (Figure 5). At each resolution, fractional area coverages of PFTs and bare
soil are determined based on the base map and written into the surface dataset as CLM4.5 inputs
(figures 5, S1, and S2). The CLM4.5 domain is terrain-following by treating the land surface as
the top of the subsurface domain, which is hydrologically active to a depth of 3.8 m. The
topography of the domain is retrieved from the 1-m topography and bathymetry dataset
[*Coleman et al.*, 2010] based on the North American Vertical Datum of 1988 (NAD88) and
resampled to each resolution (Figure S3).
The simulations were driven by hourly meteorological forcing from the Hanford
meteorological stations and hourly river stage from the gaging station over the period of 2009-
2015. Precipitation, wind speed, air temperature, and relative humidity were taken from the 300
Area meteorological station (longitude 119.726$^{o}$, latitude 46.578$^{o}$), located ~1.5 km from the
modeling domain. Other meteorological variables, such as downward shortwave and longwave
radiation, were obtained from the Hanford Meteorological station (longitude 119.599°, latitude
46.563°) located in the center of the Hanford site. The first two years of simulations (i.e., 2009
and 2010) were discarded as the spin-up period, so that 2011-2015 is treated as the simulation
period in the analyses.
Among the hydro-climatic forcing variables (e.g., river stage, surface air temperature,
incoming shortwave radiation, and total precipitation), river stage displayed the greatest inter-
annual variability (Figure 6). During the study period, high river stages occurred in early summer
of 2011 and 2012 due to the melt of above-average winter snow packs in the upstream drainage
basin, typical flow conditions occurred in 2013 and 2014, while 2015 was a year with low
upstream snow accumulation. Meanwhile, the meteorological variables, especially temperature
and shortwave radiation, do not show much inter-annual variability or trend, while precipitation
in late spring (i.e., May) of 2012 is higher than that in the other years, coincident with the high
river stage in 2012. In the "elevated" experiments (i.e., $S_{E2m}$, $S_{E10m}$, and $S_{E20m}$), the observed
river stage (meters based on NAD88) was increased by five meters at each hourly time step to
mimic a perturbed hydro-climatic condition in response to future warming.
To evaluate effects of river water and groundwater exchanges on land surface energy
partitioning, we separated the study domain for the 2-m simulations with lateral water exchange
(i.e., $S_{2m}$ and $S_{E2m}$) into two sub-domains based on 2-m topography (shown in Figure S3a): (a)
the inland domain where the surface elevation is higher than 110 m; and (b) the riparian zone
where the surface elevation is less than or equal to 110 m. In addition to the latent heat flux, the
evaporative fraction, defined as the ratio of the latent heat flux to the sum of latent and sensible
heat fluxes was calculated over the sub-domains for both observed and elevated conditions at a
daily time step for all days with significant energy inputs (i.e., when net radiation is greater than
50 W m$^2$). The evaporative is an indicator of the type of surface as summarized in literature
[*Lewis*, 1995]: it is typically less than one over surfaces with abundant water supplies, ranges
between 0.75-0.9, 0.5-0.7, 0.15-0.3 for tropical rainforests, temperate forests and grasslands,
semi-arid landscapes, respectively, and approaches 0 over deserts.
To better quantify the spatio-temporal dynamics of stream-aquifer interactions, a
conservative tracer with a mole fraction of one was applied at the river boundary to track the flux
of river water and its total mass in the subsurface domain. While a constant concentration was
maintained at the river (i.e., eastern) boundary, the tracer was allowed to be transported out of
the northern, western, and southern boundaries. Water infiltrating at the upper boundary based on
CLM4.5 simulations was set to be tracer free, while a zero-flux tracer boundary condition was
applied at the lower boundary. The initial flow condition was a hydrostatic pressure distribution
based on the water table, as interpolated from the same set of wells that were used to create the
transient lateral flow boundary conditions at the northern, western, and southern boundaries. The
initial conservative tracer concentration was set to be zero for all mesh elements in the domain.
The simulations were started on 1 January 2009 and the first two years were discarded as the
spin-up period in the analysis. The mass of tracers in the domain and the fluxes of tracers across
the boundary allow us to quantitatively understand how river water is retained and transported in
the subsurface domain.
A standalone CLM4.5 simulation was also configured and performed (i.e., $CLM_{2m}$ in Table
1). $CLM_{2m}$ shared the same subsurface properties and initial conditions as the CLM4.5 setup in
$S_{2m}$ and $S_{v2m}$ where CP v1.0 were used. However, we note that $CLM_{2m}$ are not directly
comparable to other simulations listed in Table 1 for following reasons: (1) The CLM4.5
simulates subsurface hydrologic processes only up to 3.8 m below the surface, while in the CP
v1.0 subsurface domain extends up to ~30 m below the surface; (2) as discussed in section 2.1,
CLM4.5 uses TOPMODEL-based parameterizations to simulate surface and subsurface runoffs,
as well as mean groundwater table depth using formulations derived from catchment hydrology
that are only applicable at coarser resolutions; (3) The key hydrologic processes (i.e., the
exchange of river water and groundwater at the east boundary and lateral transfer of water at all
other boundaries) that affect the hydrologic budget of the system are missing from CLM4.5.
Therefore, the simulated latent heat fluxes from $CLM_{2m}$ are only provided as a reference for
interested readers in Figure S4 and were not analyzed in section 4.

## 4    Results

### 4.1    Model evaluation

For the 3-D numerical experiments driven by the observed river stage time series (i.e., $S_{2m}$,
$S_{10m}$, $S_{20m}$),   CP v1.0 simulated soil water pressure was converted to water table depth and

compared against observed values at selected wells that were distributed throughout the domain and of variable distances from the river (Figures 7, S5 and Table 3). The model performed very well in simulating the temporal dynamics of the water table at all resolutions. The root-mean-square errors were 0.028 m, 0.028 m, and 0.023 m at 2-m, 10-m, and 20-m resolutions, respectively. The corresponding Nash–Sutcliffe coefficients were 0.998, 0.998, and 0.999. It was surprising that the performance metrics at 20-m resolution matches the observations better than those at finer resolutions, but the differences were marginal given the close match between the model simulation results and observations. River stage was clearly the dominant driving factor for water table fluctuations at the inland wells. In addition, errors in water and tracer budget conservations, and surface energy conservation for each time step in $S_{2m}$ are shown in figures S6a, b, and c respectively. The errors are sufficiently small when compared to the magnitudes of the related fluxes to ensure faithful simulations in CP v1.0. These results indicated that the coupled model was capable of simulating dynamic stream-aquifer interactions in the near shore groundwater aquifer that experiences pressure changes induced by river stage variations at sub-daily time scales.Effect of stream-aquifer interactions on land surface energy partitioning

Next we evaluated the role of water table fluctuations on land surface variables, including latent heat (LH) and sensible heat (SH) fluxes. The site is characterized by an approximate 10 m vadose zone and surface fluxes and groundwater dynamics are typically decoupled [*Maxwell and Kollet*, 2008], especially over the inland portion of the domain covered by shallow-rooted PFTs and with higher surface elevations. However, river discharge and water table elevation displayed large seasonal and inter-annual variability in the study period. Therefore, we selected the month of June in each year to assess potential land surface-groundwater coupling because it is the month of peak river stage, while energy input is high and relatively constant across the years (Figure 8a).

In June 2011 and 2012, high river stages push the groundwater table to ~108 m (or ~6 m below the land surface). Groundwater at that elevation can affect land surface water and energy exchanges with the atmosphere. The shrubs, including the patch of Basin big sagebrush and the mixture of rabbitbrush and bunchgrass on the slope close to the river, are able to tap into the elevated water table with their deeper roots. In the inland portion of the domain, capillary supply was most evident in high-water years (i.e., 2011 and 2012), remains influential in normal years (i.e., 2013 and 2014), and is essentially disabled in low-water years (i.e., 2015). The lateral

discharge of shallow groundwater to the river led to a band of negative difference in LH between
$S_{2m}$ and $S_{v2m}$ at the river boundary when the stage was low due to a decrease in rooting zone soil
moisture for evapotranspiration by the riparian trees (Figure 8b). This pattern was most evident
in June 2015. Such a mechanism decreases in high-water and normal years because of more
frequent inundation of the river bank and groundwater gradient reversal.

469        Driven by elevated river stages, land surface energy partitioning in $S_{E2m}$ (figures 9 and 10)

was significantly shifted from that in $S_{2m}$ (Figure 8a) through two mechanisms: (1) expanding
the periodically inundated fraction of the riparian zone (i.e., surface elevation ≤ 110 m); and (2)
enhancing moisture availability in the vadose zone in the inland domain (i.e., surface elevation >
110 m) through capillary rise. Both mechanisms led to general increases in simulated vadose-
zone moisture availability and therefore higher latent heat fluxes compared to the simulations
driven by the observed condition. For the inland domain, evaporative fraction clearly displayed
an increasing trend as the groundwater table level becomes shallower, consistent between the
simulations (Figure 10c). The daily evaporative fractions for the inland domain stayed well
below 0.2 when the water table levels are less than 112 m, suggesting decoupled surface-
subsurface conditions in a typical semi-arid environment. When water table levels increased to
be above 112 m, the evaporative fraction increases to ~0.2, indicating that the surface and
subsurface processes become more strongly coupled because of improved water availability for
evapotranspiration, especially in the elevated simulation (i.e., $S_{E2m}$). Evaporative fraction in the
riparian zone remained close to 1.0, suggesting strong influences of the river and the role of
deeper rooted plant types (e.g., riparian trees and shrubs) in modulating the energy partitioning
(Figure 10d) of riparian zones in the semi-arid to arid environments.

486        To confirm the above findings, the liquid saturation [*unitless*] and mass of river water [*mol*]

in the domain from $S_{2m}$ and $S_{E2m}$ on 30 June each year are plotted along a transect perpendicular
to the river ($y = 200$ m) in figures 11 and S7, and across a x-y plane at an elevation of 107 m in
figures S8 and S9, respectively. Driven by the pressure introduced by elevated river stages, river
water not only intruded further toward or even across the western boundary in high water years,
but also led to shallower water table and increased liquid saturation in the vadose zone due to
capillary rise across the domain. In fact, liquid saturation in the shallow vadose zone could
increase from 0.1-0.2 in $S_{2m}$ to 0.3-0.4 in $S_{E2m}$ on these days because of river water intrusion.
The river-water tracer could show up in the near-surface vadose zone at a distance of ~400 m
from the river (Figure S7). Interestingly, by comparing the spatial distributions of river-water
tracer in the low-water year (i.e., 2015) between the "observed" and "elevated" scenarios, the
presence of river water in the domain was much less in the elevated scenario in terms of its
spatial coverage (figures 11 and S7). This pattern suggests that after a number of years of
enhanced river water intrusion into the domain, the hydraulic gradient between groundwater and
river-water could be reversed, so that groundwater discharging might be expected more
frequently in low-water years in a prolonged elevated scenario.
The responses of LH and evaporative fraction (figures 9 and 10) indicated that a tight
coupling among stream, aquifer, and land surface processes occurred in the elevated scenario,
which could become realistic in one to two decades for the study site, or for other sites along the
Hanford reach characterized by lower elevations under the current condition.

## 4.2   Effect of spatial resolution

To apply the model to large-scale simulations or over a long time period, it is important to assess
how the model performs at coarser resolution, as the 2-m simulations are computationally
expensive. Here, we use the 2-m simulations (i.e., $S_{2m}$ and $S_{E2m}$) simulations as benchmarks for
this assessment. That is, $S_{2m}$ and $S_{E2m}$ simulated variables are treated as the "truth" for
"observed" and "elevated" river stage scenarios, and outputs from other simulations are
compared to them to verify their performance. In the previous section, we showed that simulated
water table levels from the model were virtually identical to observations. In this section, we
further quantify biases of other variables of interest from the high-fidelity 2-m simulations.
The domain-averaged daily surface energy fluxes from $S_{2m}$ show clear seasonal patterns,
which are consistent in terms of their magnitudes and timing, reflecting mean climate conditions
at the site (Figure S10). Driven by elevated river stages, latent heat from $S_{E2m}$ is consistently
higher than that from $S_{2m}$. The mean latent heat and sensible heat fluxes simulated by $S_{2m}$ were
14.1 W m$^{-2}$ and 38.7 W m$^{-2}$ over this period, compared to by 18.50 W m$^{-2}$ and 35.75 W m$^{-2}$ in
$S_{E2m}$. Figure 12 shows deviations of simulated LH and SH in the 20-m and 10-m simulations
from the corresponding 2-m simulations. The deviations of both LH and SH were small across
all the simulations driven by the observed river stage when surface and subsurface were
decoupled. In the elevated simulations (i.e., $S_{E10m}$ and $S_{E20m}$) when surface and subsurface
processes are more tightly coupled, errors in surface fluxes became significant in the coarse
resolution simulations when compared to $S_{E2m}$. For example, the relative errors in LH were
2.41% and 1.35% for $S_{20m}$ and $S_{10m}$, respectively, as compared to $S_{2m}$, but grew as large as
33.84% and 33.19% for $S_{E20m}$ and $S_{E10m}$, respectively, when compared to $S_{E2m}$. The 10-m
simulations outperformed the 20-m simulations under both scenarios but the magnitudes of
errors were comparable. On the other hand, notably the vertical only simulation ($S_{v2m}$) has a
small error of 5.67% in LH compared to $S_{2m}$, indicating that lateral flow is less important when
water table is deep.
To better understand how water in the river and the aquifer was connected, we also
quantified the biases of subsurface state variables and fluxes including total water mass and
tracer amount, as well as exchange rates of water and tracer at four boundaries of the subsurface
domain using a similar approach (Figure S11 and Figure 13). Compared to the magnitude of total
water mass in the domain (averaged 919.45 $\times 10^6$ Kg and 1020.19 $\times 10^6$ Kg in $S_{2m}$ and $S_{E2m}$),
errors introduced by coarsening the resolution were very small under the observed river stage
condition (0.04% for $S_{20m}$ and 0.03% for $S_{10m}$) and grew to 9.85% for $S_{E20m}$ and 9.87% for $S_{E10m}$
in terms of total water mass in the domain (Table 5). However, for total tracer in the domain
(averaged 142.07$\times 10^6$ mol and 172.46 $\times 10^6$ mol in $S_{2m}$ and $S_{E2m}$) as a result of transport of river
water in lateral and normal directions to the river, resolution clearly makes a difference under
both observed condition and elevated scenarios (relative errors of 5.44% for $S_{10m}$, 10.40% for
$S_{20m}$, and 22.0% for both $S_{E10m}$ and $S_{E20m}$). The magnitude of computed mass exchange rates at
the four boundaries (Figure S11) indicates that a coarse resolution promotes larger river water
fluxes and groundwater exchanges, especially during the period of spring river stage increase
under the elevated scenario. This forcing contributes to a significant bias in total tracer amount
by the end of the simulation. The exchange rates at the other three boundaries follow the same
pattern but with smaller magnitudes, especially for the west boundary that requires a significant
gradient high enough to push river water further inland.
The results of simulations at three different resolutions indicated that: (1) the partitioning of
the land surface energy budget is mainly controlled by near-surface moisture. Spatial resolution
did not seem to be a significant factor in the computation of surface energy fluxes when the
water table was deep at the semi-arid site; (2) if the surface and subsurface are tightly coupled as
in the elevated river stage simulations, resolution becomes an important factor to consider for
credible simulations of the surface fluxes, as the land surface, subsurface, and riverine processes
are expected to be more connected and coupled; (3) regardless of whether a tight coupling
between the surface and subsurface occurs, if mass exchange rates and associated
biogeochemical reactions in the aquifer are of interest, a higher resolution is desired close to the
river shoreline to minimize terrain errors.

## 561 5 Discussion and future work

A coupled three-dimensional surface and subsurface land model was developed and applied to a
site along the Columbia River to simulate interactions among river water, groundwater, and land
surface processes. The model features the coupling of the open-source and state-of-the-art
models portable on HPCs, the multi-physics reactive transport model PFLOTRAN and the
CLM4.5. Both models are under active development and testing by their respective communities,
therefore the coupled model could be updated to newer versions of PFLOTRAN and/or CLM to
facilitate transfer of knowledge in a seamless fashion. The coupled model represents a new
addition to the integrated surface and subsurface suite of models.
By applying the coupled model to a field site along the Columbia River shoreline driven by
highly dynamic river boundary conditions resulting from upstream dam operations, we
demonstrated that the model can be used to advance mechanistic understanding of stream-
aquifer-land interactions surrounding near-shore alluvial aquifers that experience pressure
changes induced by river stage variations along  managed river reaches, which are of global
significance as a result of over 30,000 dams constructed worldwide during the past half century.
The land surface, subsurface, and riverine processes along such managed river corridors are
expected to be more strongly coupled under projected hydro-climatic regimes as a result of
increases in winter precipitation and early snowmelt. The dataset presented in this study can
serve as a good benchmarking case for testing other coupled models for their applications to such
systems. More data needs to be collected to facilitate the application and validation of the model
to a larger domain for understanding the contribution of near-shore hydrologic exchange to water
retention, biogeochemical cycling, and ecosystem functions along the river corridors.
By benchmarking the coarser resolution simulations at 20 m and 10 m against the 2-m
simulations, we find that resolution is not a significant factor for surface flux simulations when
the water table is deep. However, resolution becomes important when the surface and subsurface
processes are tightly coupled, and for accurately estimating the rate of mass exchange at the
riverine boundaries, which can affect the calculation of biogeochemical processes involved in
carbon and nitrogen cycles.
Our numerical experiments suggested that riverine, land surface, and subsurface processes
could become more tightly coupled through two mechanisms in the near-shore environments: (1)
expanding the periodically inundated fraction of the riparian zone and (2) enhancing moisture
availability in the vadose zone in the inland domain through capillary rise. Both mechanisms can
lead to increases in vadose-zone moisture availability and higher evapotranspiration rates. The
latter is critical for understanding ecosystem functioning, biogeochemical cycling, and land-
atmosphere interactions along river corridors in arid and semi-arid regions that are expected to
experience new hydro-climatic regimes in a changing climate. However, these systems have
been poorly accounted for in current-generation Earth system models and therefore require more
attention in future studies.
We acknowledge that there are a number of limitations of this study that need to be addressed
in future studies:
(1) Motivated by understanding the stream-aquifer-land interactions with a focus on
groundwater and river water interactions along a river corridor situated in a semi-arid climate,
the river boundary conditions were prescribed using observations with gaps filled by a 1-D
hydrodynamics model. Future versions of the CP model need to incorporate two-way
interactions between stream and aquifer by developing a surface flow component and testing the
new implementation against standard benchmark cases [*Kollet et al*., 2017; *Maxwell et al*.,
2014].
(2) We note that CLM estimates the surface heat and moisture fluxes using the Monin-
Obukhov Similarity Theory (section 2.1), which is only valid when the surface layer depth $z \gg z_0$,
where $z_0$ is the aerodynamic roughness length. As reviewed by *Basu and Lacser* [2017], it is
highly recommended that $z > 50z_0$, which should be proportional to the horizontal grid spacing to
guarantee the validity of the Monin-Obukhov Similarity Theory [*Arnqvist and Bergström*, 2015].
In our simulations, the majority of the Hanford 300A domain is covered by bare soil ($z_0 = 0.01$
m), grass ($z_0 = 0.013$ m), shrubs ($z_0 = 0.026$-$0.043$ m), and riparian trees (varies across the
seasons, $z_0 = 0.008$ m when LAI = 2 in the summer and $z_0 = 1.4$ when LAI = 0 in the winter).
Therefore, a 2-m resolution is sufficiently coarse under most conditions except for the grid cells
covered by riparian trees in the winter. Nevertheless, the wintertime latent heat and sensible heat
fluxes are nearly zero due to extremely low energy inputs. Therefore, the 2-m simulations
supported by the dense groundwater monitoring network at the site provide a valid benchmark
for the coarser resolution simulations. For future applications of the coupled model, caution
should be taken to evaluate the site condition for the validity of model parameterizations.
(3) We used the simulated surface energy fluxes from $S_{2m}$ to verify coarser-resolution
simulations. The simulated surface energy flux needs to be validated against eddy covariance
tower observations, which are not available yet at the site. Nevertheless, we have made initial
efforts to install eddy covariance systems at the site (see description in section 3.1 of *Gao et al.*
[2017]) but the processing the flux data is still preliminary. We will report flux observations and
validations of the surface energy budget simulations in future studies.
(4) Even when observed fluxes are available for validation, the model structural problems
associated with ET parameterizations in CLM4.5 need to be addressed for reasonable
simulations of the ET components, especially for the study site. That is, it has been well-
documented that ET simulated by CLM4.5 and CLM4 could be enhanced when vegetation is
removed. This ET enhancement over bare soil has been documented as a counter-intuitive bias
for most unsaturated soils in CLM4 and CLM4.5 simulations [*Lawrence et al.*, 2012; *Tang and*
*Riley*, 2013a]. *Tang and Riley* [2013a] explored a few potential causes for this likely bias (e.g.,
soil resistance, litter layer resistance, and numerical time step). They found the implementation
of a physically based soil resistance lowered the bias slightly, but concluded that the bias
remained [*Tang and Riley*, 2013b]. Meanwhile, in studying ET over semiarid regions, *Swenson*
*and Lawrence* [2014] proposed another soil resistance formulation to fix this excessive soil
evaporation problem within CLM4.5. While their modification improved the simulated terrestrial
water storage anomaly and ET when compared to GRACE data and FLUXNET-MTE data,
respectively, the empirical nature of the soil resistance proposed could have underestimated the
soil resistance variability when compared to other estimates [*Tang and Riley*, 2013b].

**Code availability**

CLM4.5 is an open-source software released as part of the Community Earth System Model (CESM) version 1.2 (http://www.cesm.ucar.edu/models/cesm1.2). The version of CLM4.5 used in CP v1.0 is a branch from the CLM developer's repository. Its functionality is scientifically consistent with descriptions in *Oleson et al.* [2013] with source codes refactored for a modular code design. Additional minor code modifications were added by the authors to support coupling with PFLOTRAN. Permission from the CESM Land Model Working Group has been obtained to release this CLM4.5 development branch but the National Center for Atmospheric Research cannot provide technical support for this version of the code CP v1.0. PFLOTRAN is an open-source software distributed under the terms of the GNU Lesser General Public License as published by the Free Software Foundation either version 2.1 of the License, or any later version. The CP v1.0 has two separate, open-source repositories for CLM4.5 and PFLOTRAN at:

- https://bitbucket.org/clm_pflotran/clm-pflotran-trunk
- https://bitbucket.org/clm_pflotran/pflotran-clm-trunk

The README guide for the CP v1.0 and dataset used in this study are available from the open-source repository https://bitbucket.org/pnnl_sbr_sfa/notes-for-gmd-2017-35.

**Acknowledgement**
This research was supported by the U.S. Department of Energy (DOE), Office of Biological and
Environmental Research (BER), as part of BER's Subsurface Biogeochemical Research Program
(SBR). This contribution originates from the SBR Scientific Focus Area (SFA) at the Pacific
Northwest National Laboratory (PNNL).

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

**Tables and Figures**

Table 1. Summary of numerical experiments

| Experiments | Model | Horizontal Resolution | Lateral flow | River Stage (m) |
|---|---|---|---|---|
| $Sv_{2m}$ | CP v1.0 | 2m | No | Observed |
| $S_{2m}$ | CP v1.0 | 2m | Yes | Observed |
| $S_{10m}$ | CP v1.0 | 10m | Yes | Observed |
| $S_{20m}$ | CP v1.0 | 20m | Yes | Observed |
| $S_{E2m}$ | CP v1.0 | 2m | Yes | Observed +5 |
| $S_{E10m}$ | CP v1.0 | 10m | Yes | Observed +5 |
| $S_{E20m}$ | CP v1.0 | 20m | Yes | Observed +5 |
| $CLM_{2m}$ | CLM4.5 | 2m | No | Not applicable |



Table 2. Hydrogeological material properties of Hanford and Ringold materials.

| Material | Porosity | Permeability (m$^2$) | Van Genuchten/Burdine Parameters | | |
|---|---|---|---|---|---|
| | | | Res. Sat. | m | alpha |
| **Hanford** | 0.20 | $7.387 \times 10^{-9}$ | 0.16 | 0.34 | $7.27 \times 10^{-4}$ |
| **Ringold** | 0.40 | $1.055 \times 10^{-12}$ | 0.13 | 0.75 | $1.43 \times 10^{-4}$ |



Table 3.  The comparison between simulated and observed water table levels

| Well number | $S_{2m}$ | | $S_{10m}$ | | $S_{20m}$ | |
|---|---|---|---|---|---|---|
| | RMSE (m) | N-S | RMSE (m) | N-S | RMSE (m) | N-S |
| **399-3-29** | 0.022 | 0.999 | 0.022 | 0.999 | 0.021 | 0.999 |
| **399-3-34** | 0.011 | 1.000 | 0.011 | 1.000 | 0.006 | 1.000 |
| **399-2-01** | 0.039 | 0.997 | 0.038 | 0.997 | 0.029 | 0.998 |
| **399-1-60** | 0.016 | 1.000 | 0.016 | 0.999 | 0.013 | 1.000 |
| **399-2-33** | 0.028 | 0.998 | 0.028 | 0.998 | 0.022 | 0.999 |
| **399-1-21A** | 0.023 | 0.999 | 0.023 | 0.999 | 0.020 | 0.999 |
| **399-2-03** | 0.037 | 0.997 | 0.037 | 0.997 | 0.029 | 0.998 |
| **399-2-02** | 0.045 | 0.995 | 0.045 | 0.995 | 0.042 | 0.996 |
| **mean** | 0.028 | 0.998 | 0.028 | 0.998 | 0.023 | 0.999 |



Table 4. The relative error in surface energy fluxes simulated by $S_{10m}$ and $S_{20m}$ benchmarked against $S_{2m}$
and by $S_{E10m}$ and $S_{E20m}$ benchmarked against $S_{E2m}$

| Simulation | Latent heat flux (%) | Sensible heat flux (%) |
|---|---|---|
| $S_{v2m}$ | 5.67 | 1.63 |
| $S_{10m}$ | 1.35 | 0.78 |
| $S_{20m}$ | 2.41 | 1.42 |
| $S_{E10m}$ | 33.19 | 13.71 |
| $S_{E20m}$ | 33.84 | 14.18 |



Table 5. The relative error in total water mass and tracer amount in the subsurface simulated in $S_{10m}$ and
$S_{20m}$ benchmarked against $S_{2m}$ and by $S_{E10m}$ and $S_{E20m}$ benchmarked against $S_{E2m}$

| *Simulation* | Total water mass (%) | Total tracer (%) |
|---|---|---|
| $S_{10m}$ | 0.03 | 5.44 |
| $S_{20m}$ | 0.04 | 10.40 |
| $S_{E10m}$ | 9.87 | 22.00 |
| $S_{E20m}$ | 9.85 | 22.00 |



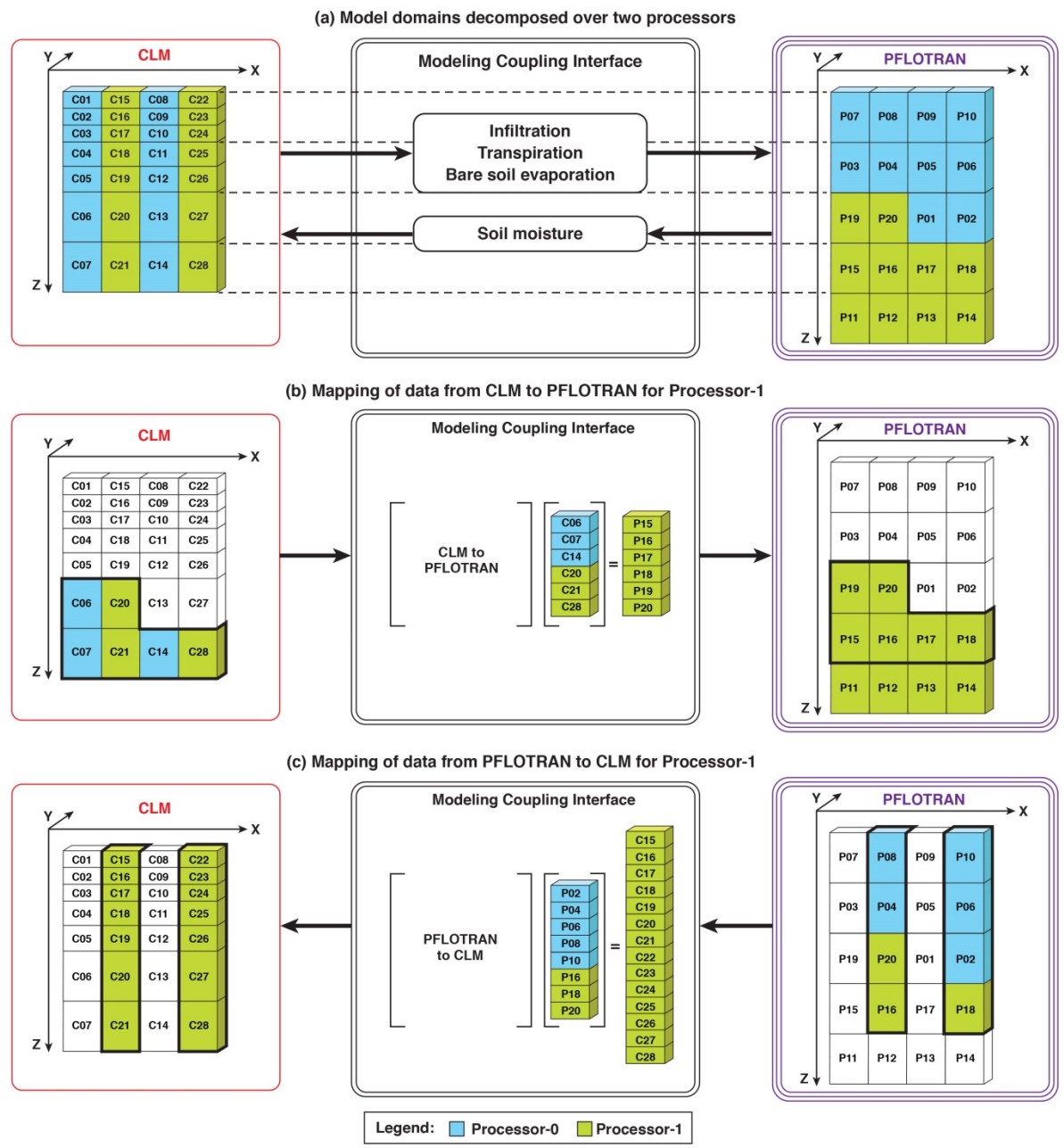

Figure 1. Schematic representations of the model coupling interface of CP v1.0. (a) Domain
decomposition of a hypothetical CLM and PFLOTRAN domain comprising of 4x1x7 and 4x1x5 grids in x,
y, and z directions across two processors as shown in blue and green. (b) Mapping of water fluxes from
CLM onto PFLOTRAN domain via a local sparse matrix vector product for grids on processor 1. (c)
Mapping of updated soil moisture from PFLOTRAN onto CLM domain via a local sparse matrix vector
product for grids on processor 1.

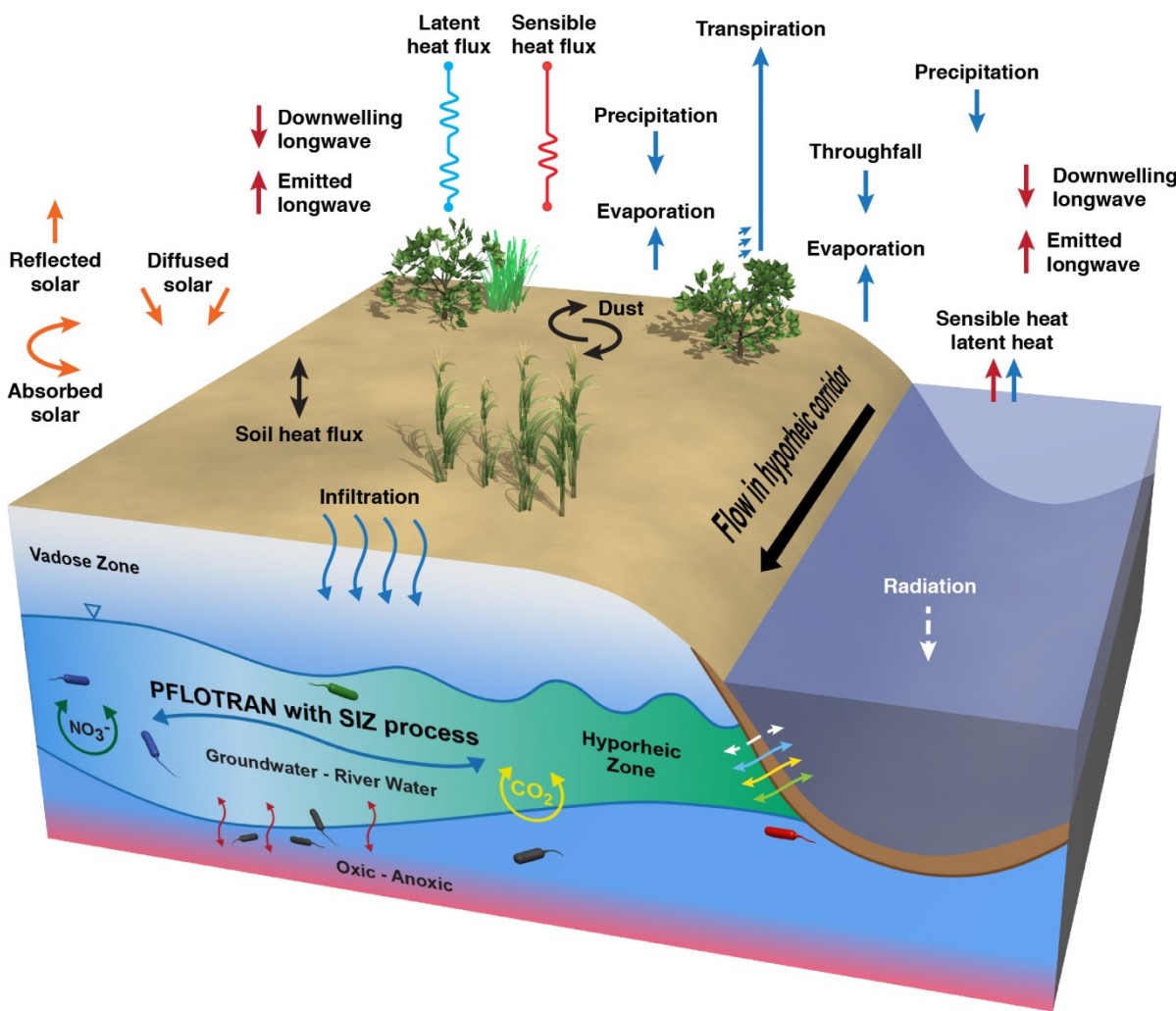

Figure 2. Schematic representation of hydrologic processes simulated in CP v1.0

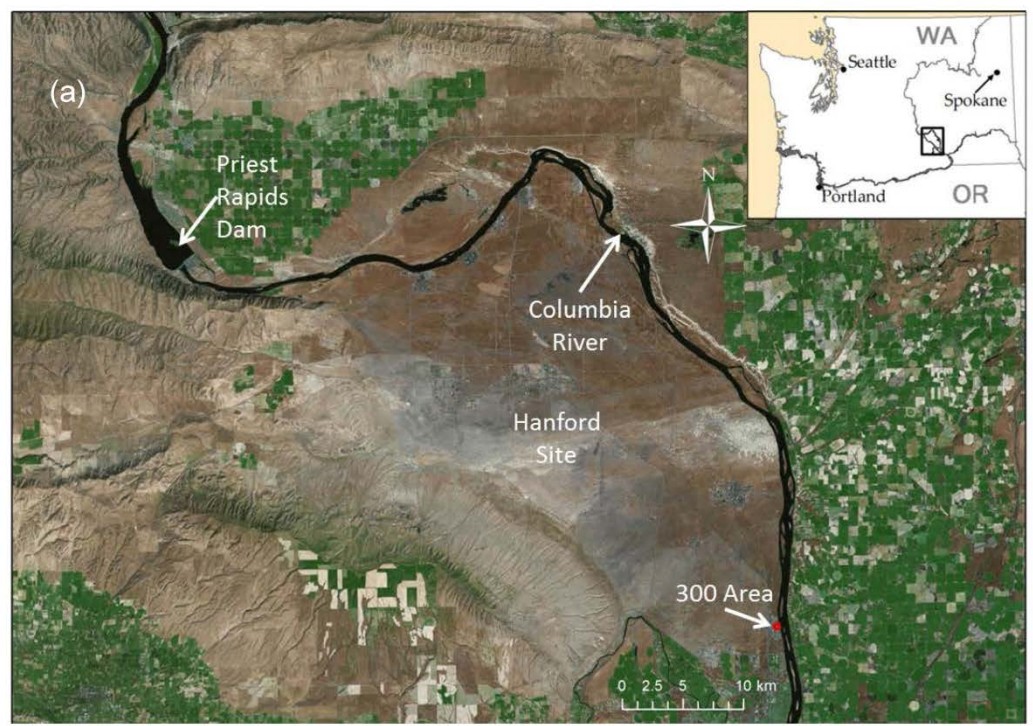

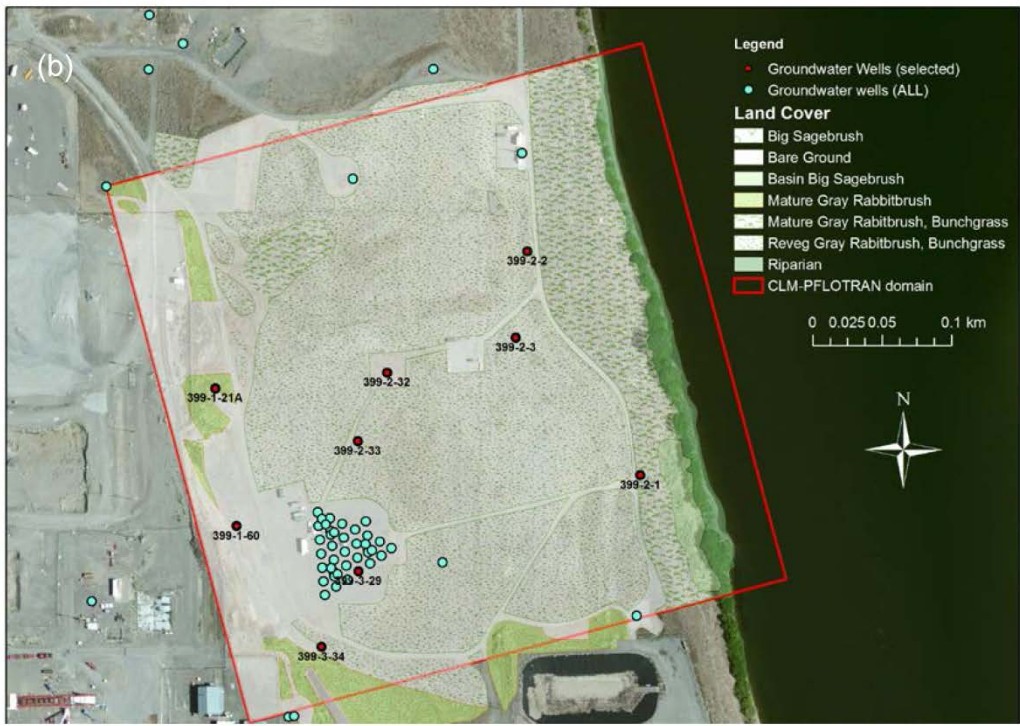


Figure 3. (a) The Hanford Reach of the Columbia River and the Hanford Site location in south-central
Washington State, USA; (b) the 400 m × 400 m modeling domain located in the Hanford 300 Area.

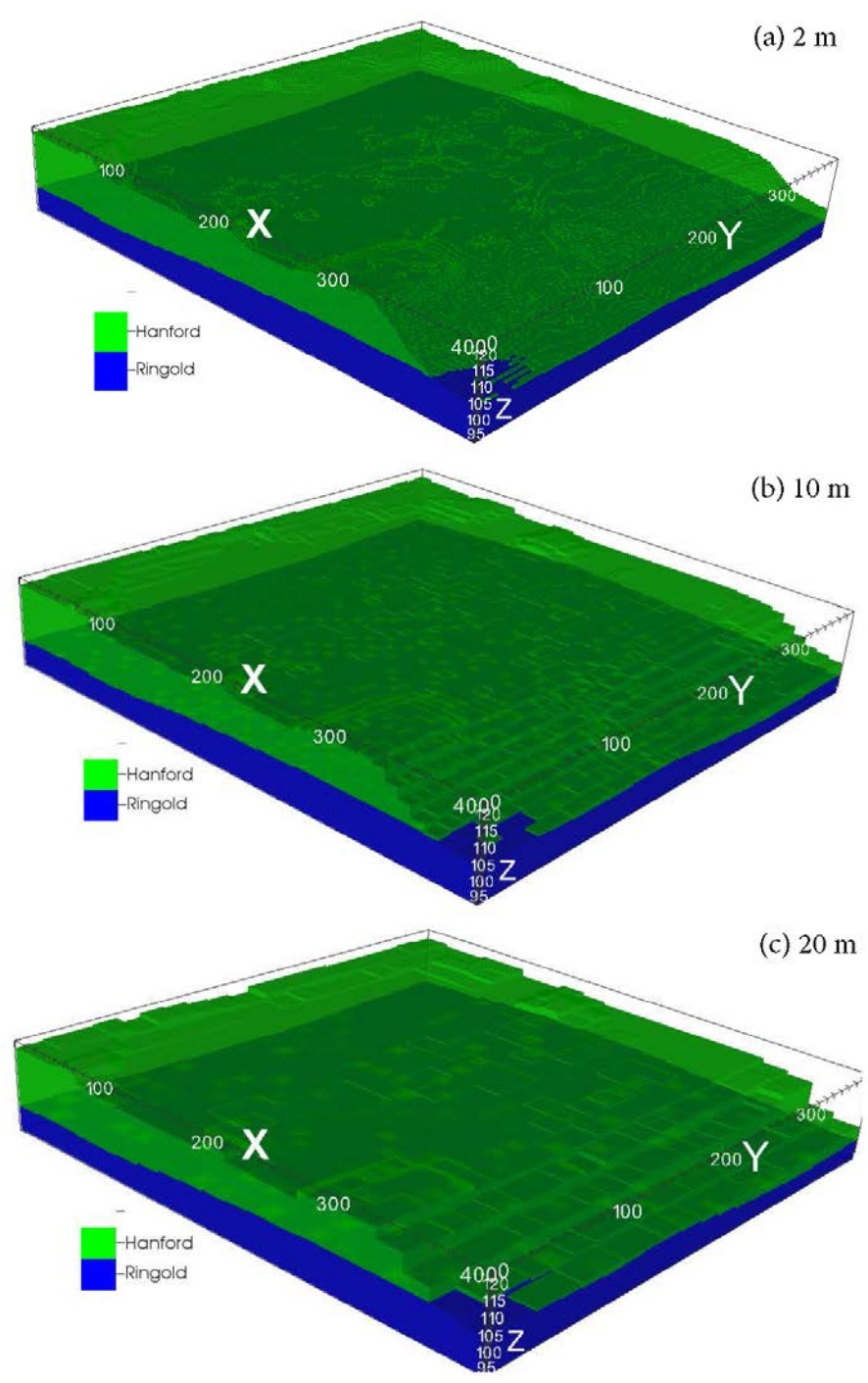


Figure 4. PFLOTRAN meshes and associated material IDs at (a) 2-m; (b) 10-m; and (c) 20-m resolutions


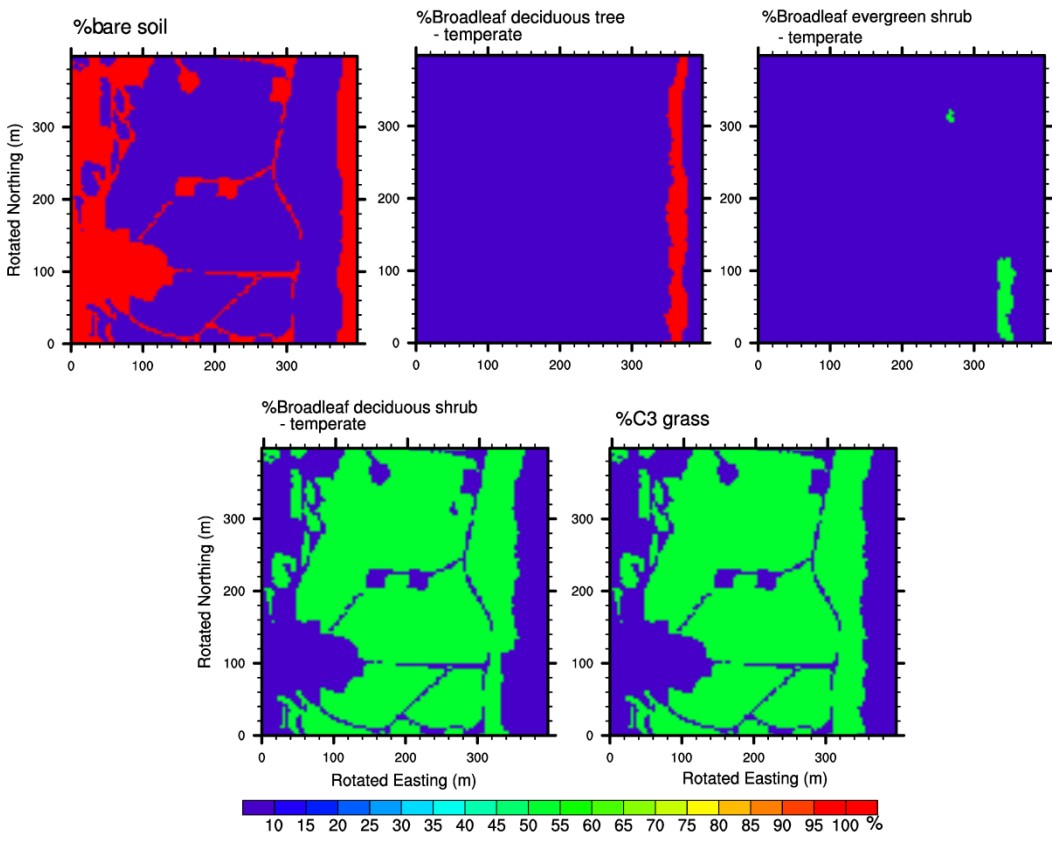


Figure 5. Plant function types at 2-m resolution as inputs for CLM4.5

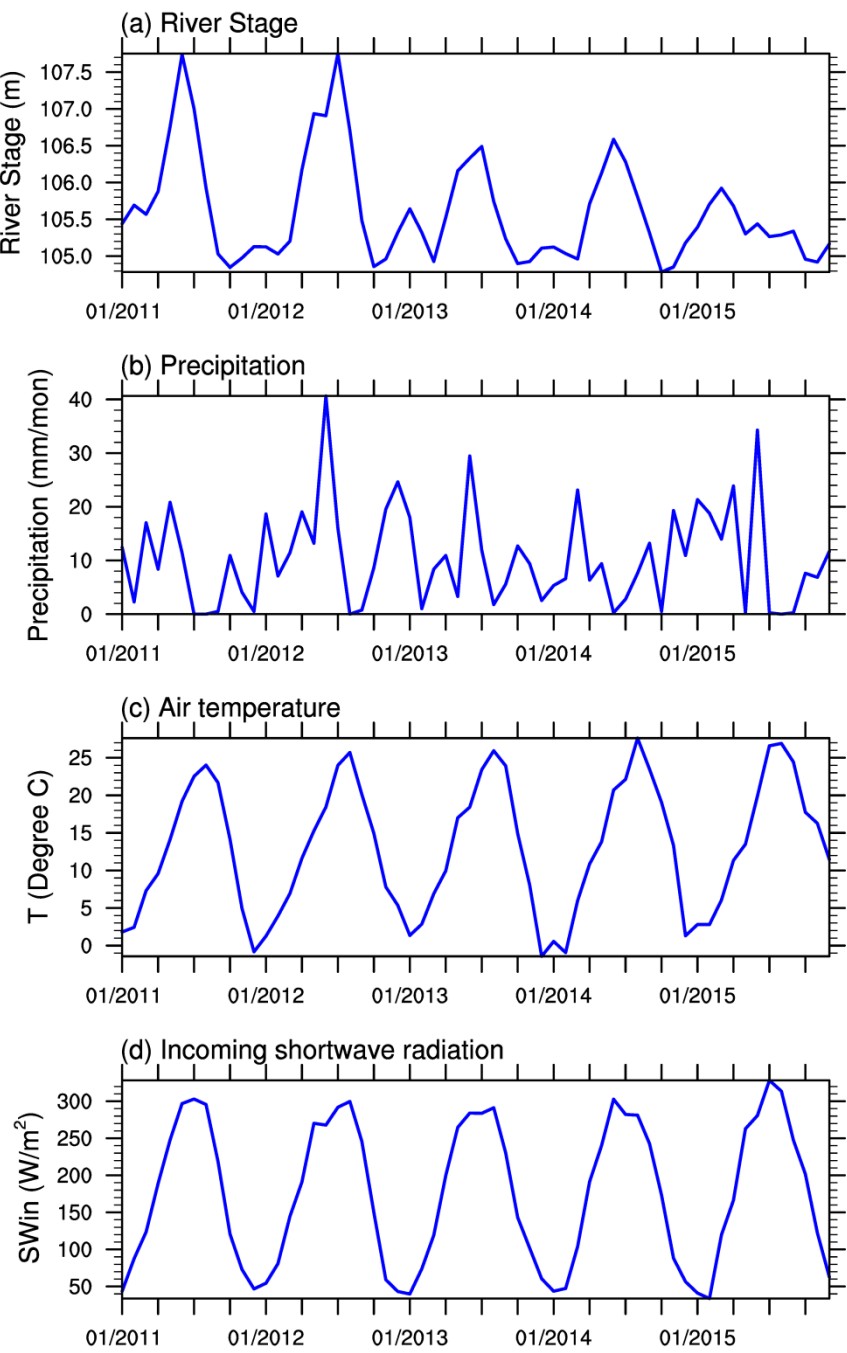


Figure 6.  Hydro-meteorological drivers in the study period: (a) monthly mean river Stage; (b) monthly
total precipitation; (c) monthly mean surface air temperature; (d) and monthly mean incoming shortwave
radiation.

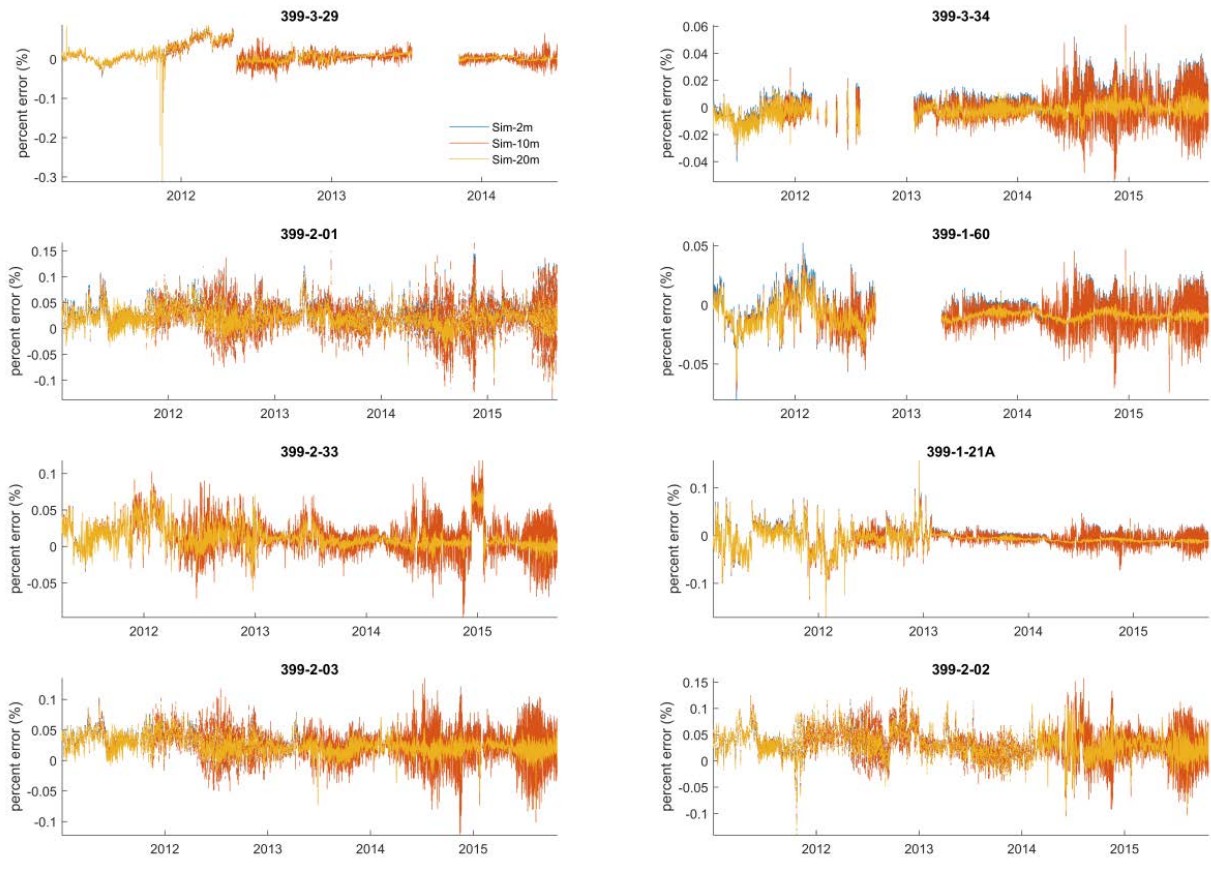


Figure 7. Deviation (in percentages) of simulated water table levels from observations at selected wells
shown in Figure 3b.

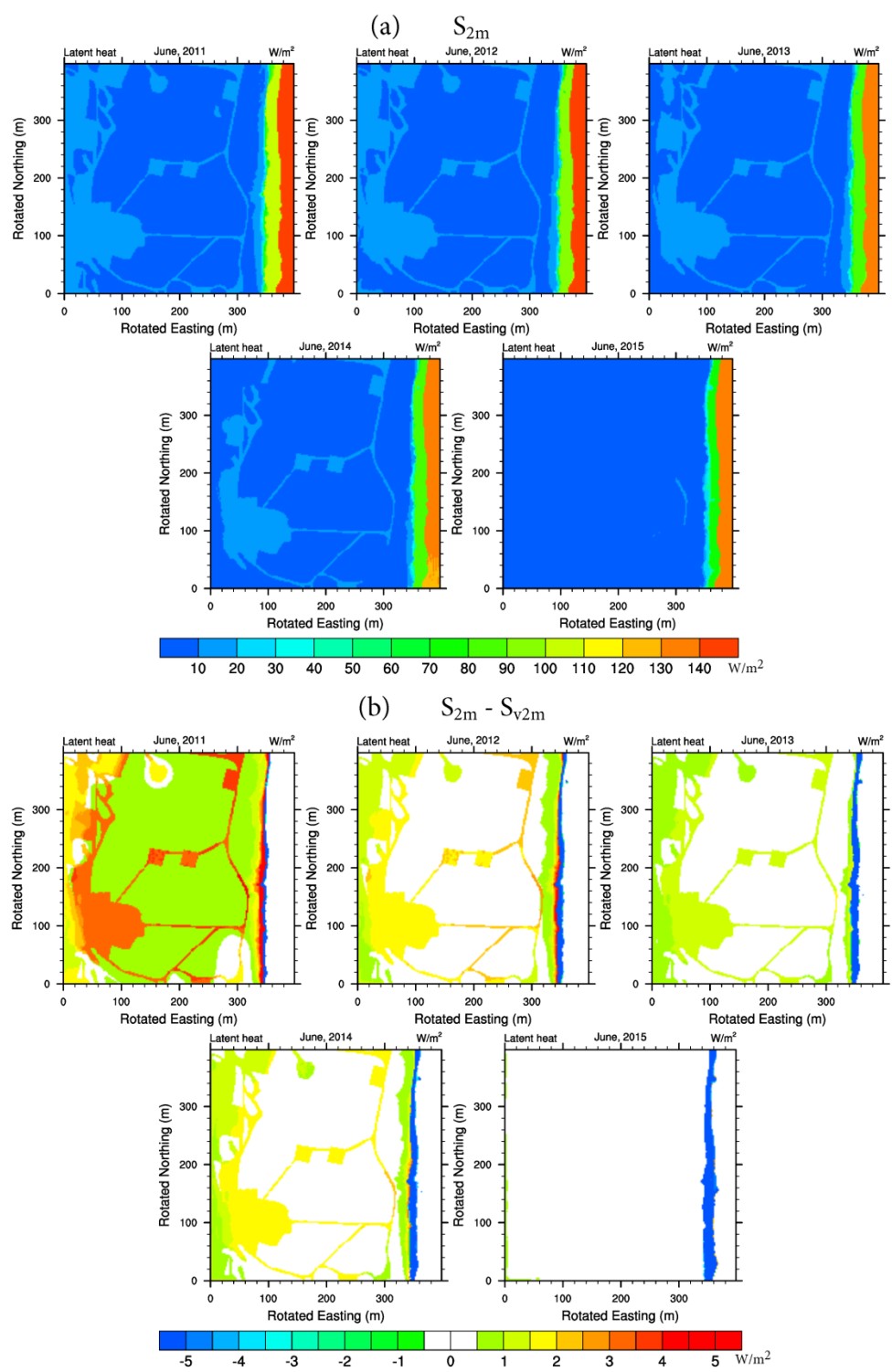


Figure 8. (a) Simulated latent heat fluxes in June from the 3-D simulation ($S_{2m}$); and (b) the difference

between the 3-D and vertical only simulations (i.e., S2m - $Sv_{2m}$).

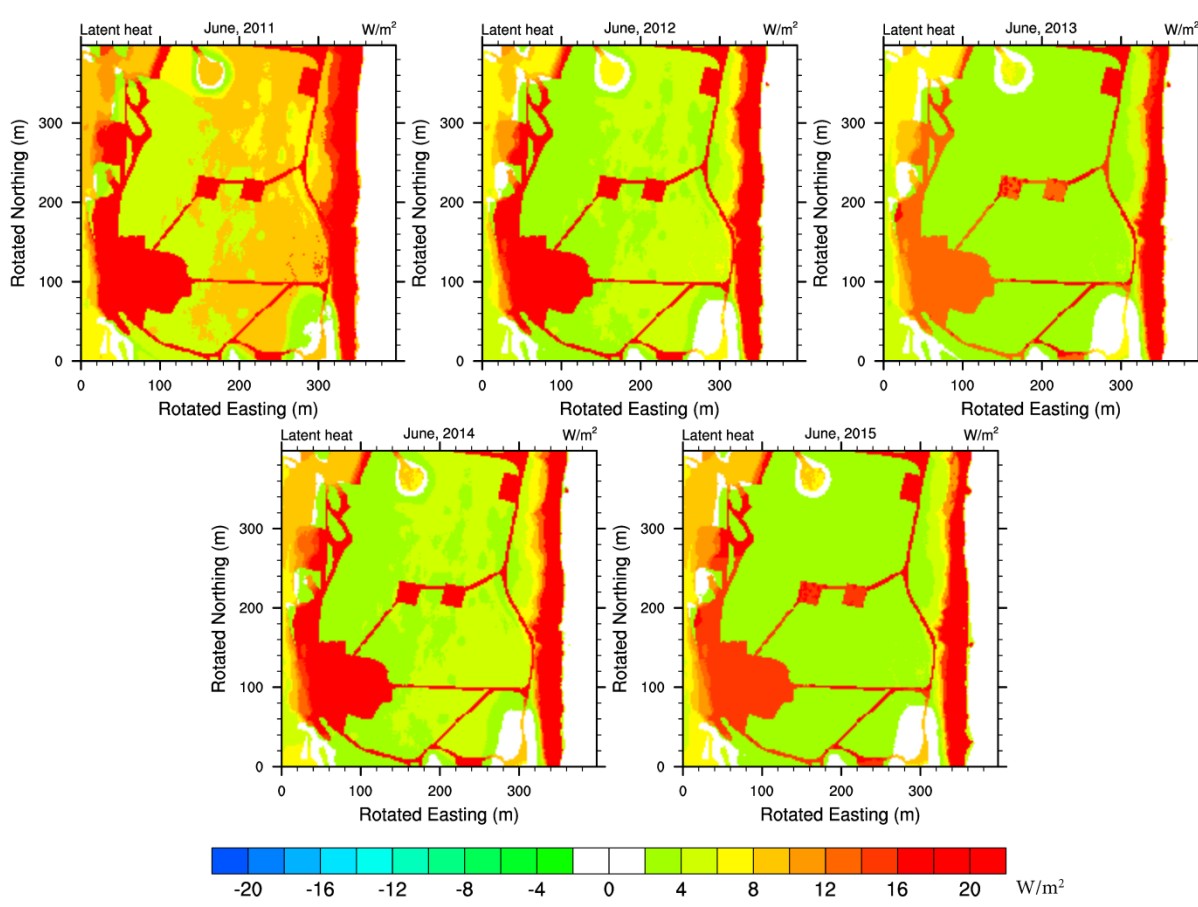


Figure 9. Difference between simulated latent heat fluxes by $S_{E2m}$ and $S_{2m}$ in June.

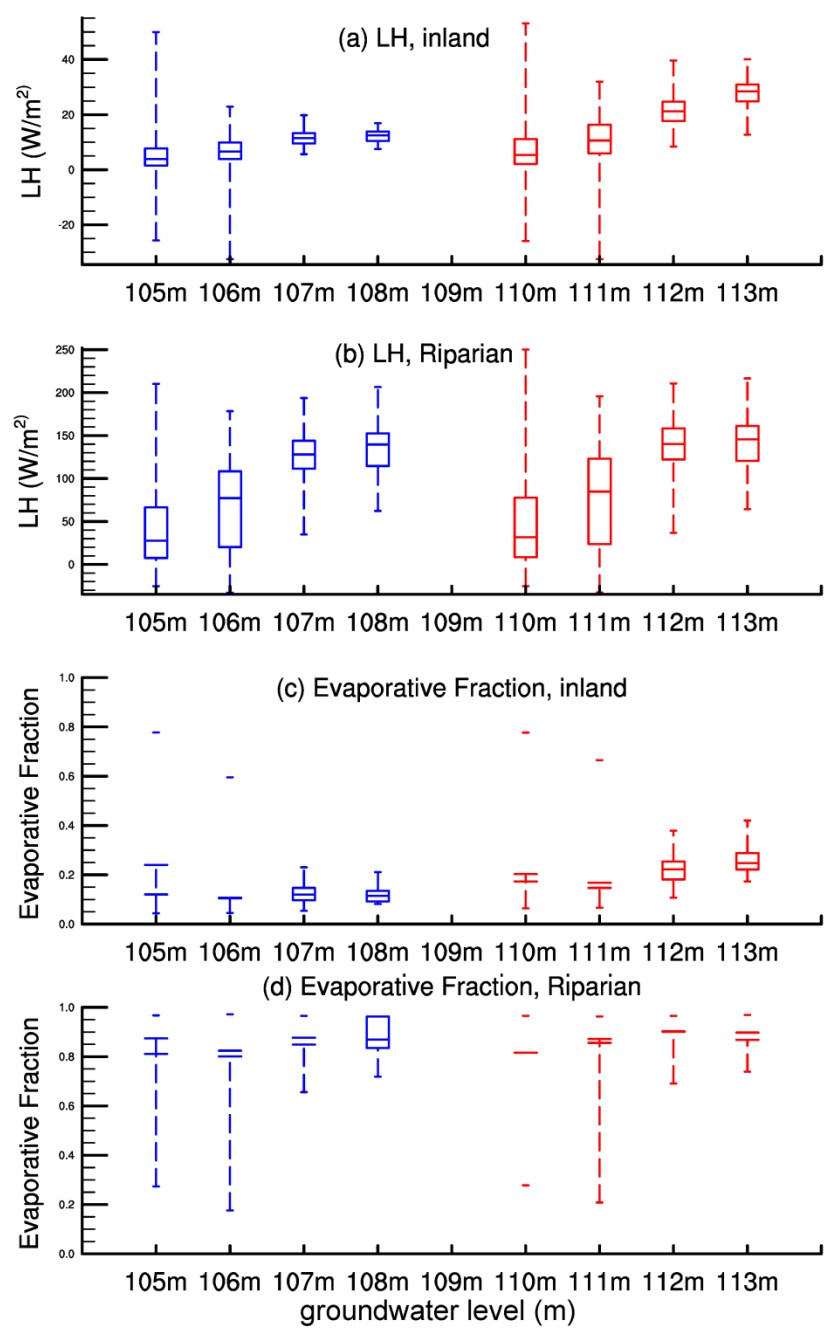


Figure 10. Boxplots of (a) land heat fluxes over the inland domain; (b) and latent heat fluxes in the
riparian zone; (c) Evaporative fractions over the inland domain; (d) Evaporative fractions in the riparian
zone in relation to groundwater table levels in the five-year period. The red boxes and whiskers represent
summary statistics from $S_{2m}$, and red ones indicate those from $S_{E2m}$. The bottom and top of each box are
the 25[th] and 75[th] percentile, the band inside the box is median, and the ends of the whiskers are
maximum and minimum values, respectively.

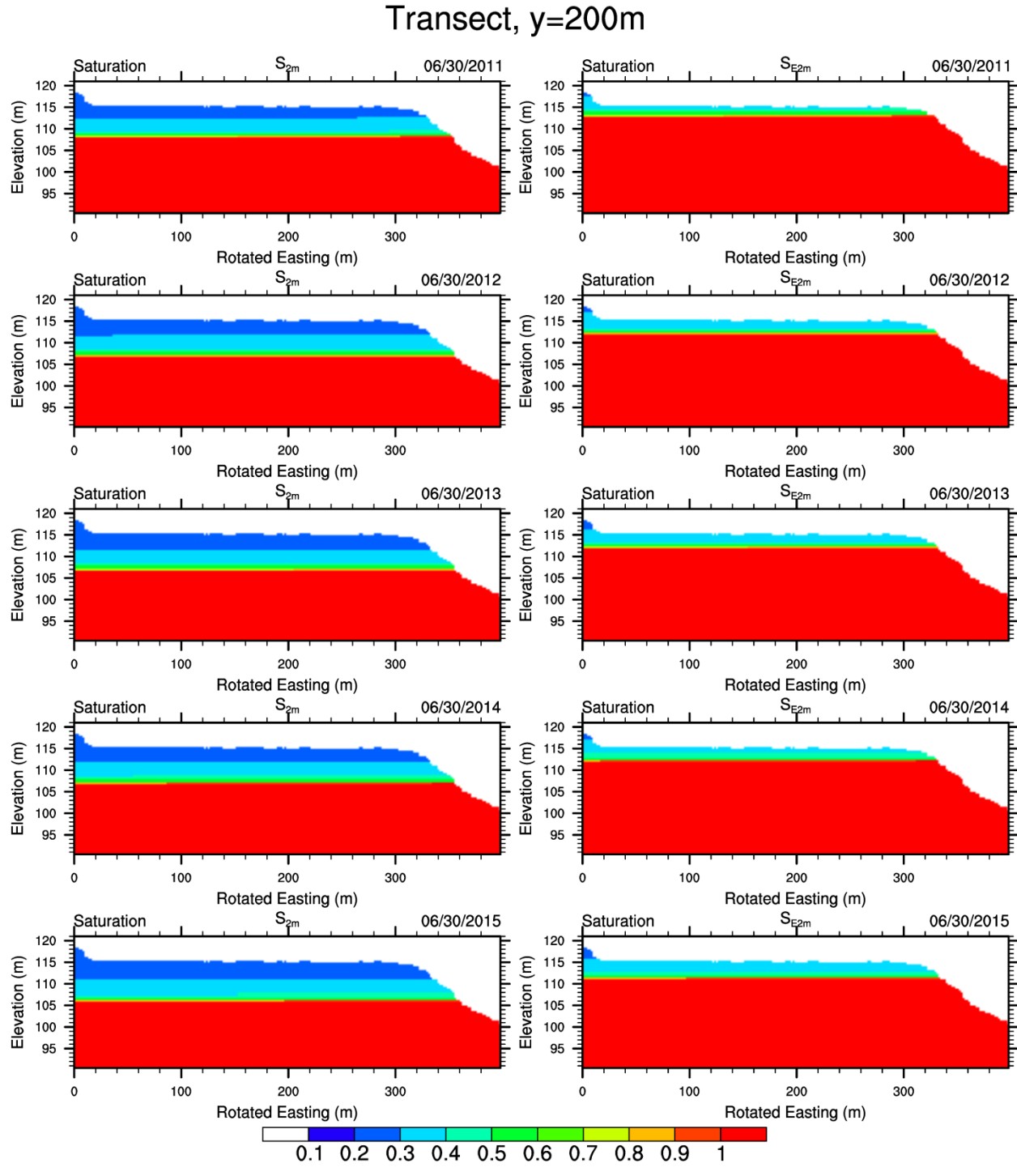


Figure 11. Liquid saturation levels (unitless) across a transect perpendicular to the river (y=200m) on 30
June of each year in the study period from (a) $S_{2m}$ and (b) $S_{E2m}$

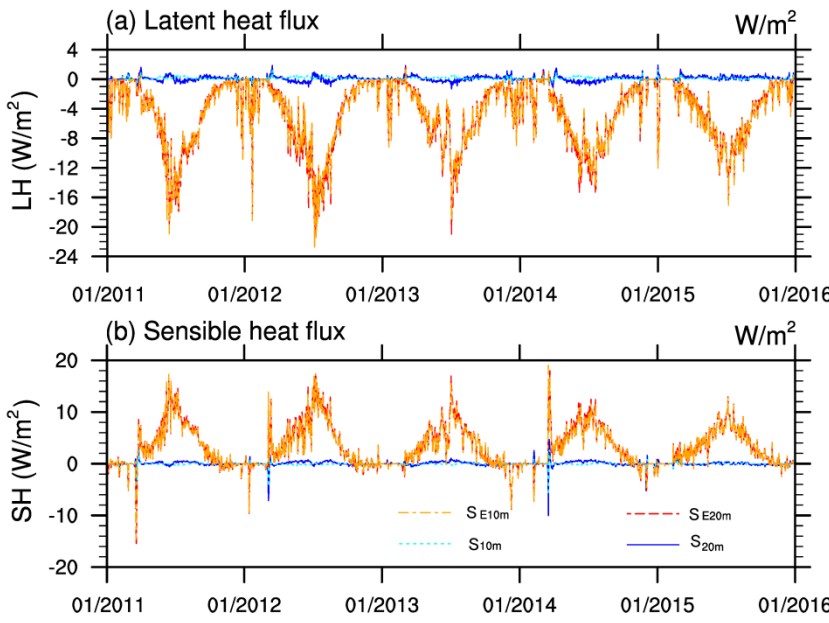


Figure 12. Deviations of simulated domain-average latent heat and sensible heat fluxes from those
simulated by $S_{2m}$ (for $S_{10m}$ and $S_{20m}$), and by $S_{E2m}$ (for $S_{E10m}$ and $S_{E20m}$).

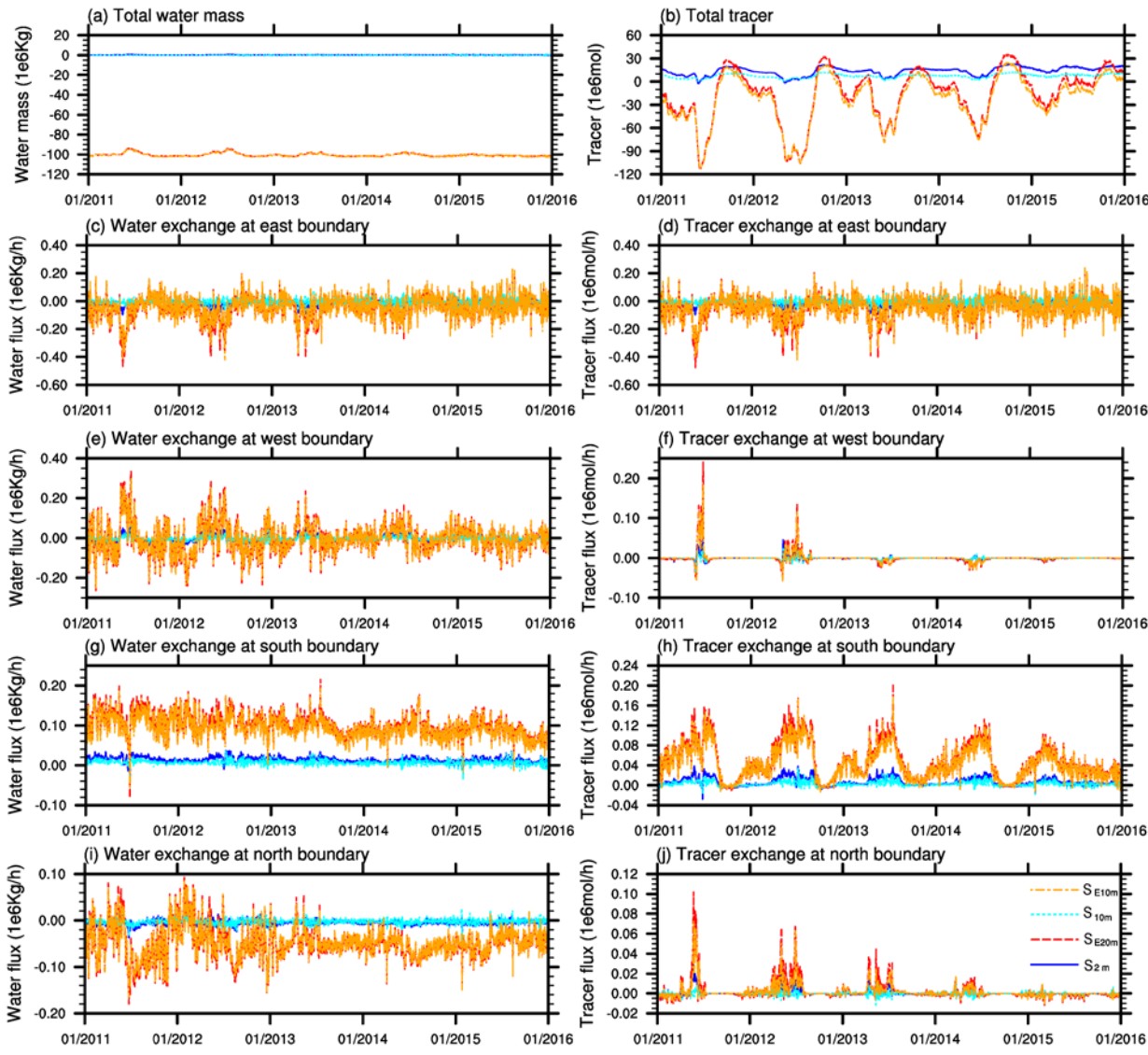

Figure 13. Deviations of total water mass, tracer, and exchange rates of water and tracer at four boundaries from
those simulated by $S_{2m}$ (for $S_{10m}$ and $S_{20m}$), and by $S_{E2m}$ (for $S_{E10m}$ and $S_{E20m}$).


