# Peer review of "Coupling a three-dimensional subsurface flow and transport model with a land surface model to simulate stream-aquifer-land interactions (CP v1.0) Gautam Bisht1, Maoyi Huang2,\*, Tian Zhou2, Xingyuan Chen2, Heng Dai2, Glenn E. Hammond3, William J. Riley1, Janelle L. Downs2, Ying Liu2, John M. Zachara2 1Lawrence Berkeley National Laboratory, Berkeley, CA 2Pacific Northwest National Laboratory, Richl"

_Geoscientific Model Development, 2017_

## Referee Comment (RC1) · Anonymous Referee #1 · 19 Apr 2017

The authors present a new coupled version of CLM4.5 and PFLOTRAN, and demonstrate the impacts of resolution, horizontal fluxes, and river stage height in simulating groundwater levels and turbulent fluxes between the land and the atmosphere. The authors demonstrate that the new model is capable of simulating the observed water table depth, independent of the model resolution.

The authors show PF-CLM results when there is no lateral subsurface exchange. Does this produce the exact same results as CLM without PFLOTRAN? If not CLM should be included in the manuscript. If so the authors should state that running PF-CLM without

horizontal transfer gives identical results to CLM.

The description of the technical details of the coupling needs more explanation. It is cleat that only the soil moisture and hydraulic properties as passed between CLM and PFLOTRAN. However how does this work given that the vertical discretization of CLM differs from PFLOTRAN? The vertical resolution of the subsurface (PFLOTRAN) component is only 0.5 meters, while CLM uses layers from mm to m. How does this impact transpiration? Is the default rooting depth used in CLM? How are the 0.5 meter thick layers mapped to the much thinner layers? Does CLM compute freezing and thawing? Which processes are no longer used by CLM in the coupled version?

I am having trouble understanding why the grasses away from the river always have near zero latent heat flux (Figure 7a) while the bare ground has a larger latent heat flux? This explains why the latent heat flux only differs over the bare soil surfaces between PFCLM2m and PFCLMv2m. I fail to understand why the bare soil has a higher latent heat flux than the vegetation, especially given that the moisture available to the roots from horizontal transfer should be even greater than the moisture at the surface. The authors need to explain if this the expected behavior in CLM, or if it is due to the coupling between PFLOTRAN and CLM.

Figure 6 should be shown as the difference between the observations and the simulations. This will show much more information concerning how the simulations differ.

---

## Referee Comment (RC2) · Anonymous Referee #2 · 3 May 2017

In this manuscript the authors document development and application of a coupled Richards' equation solver (PFLOTRAN) with a land surface model (CLM 4.5) and apply it to a test problem developed from an intensely observed floodplain system. This manuscript is generally clearly written but in my opinion needs to better articulate its contributions given the prior work on this topic. I have specific comments below that need to be addressed before the suitability of this work for GMD can be assessed. The larger comments are ones of contribution, what does this work want to contribute to our understanding of coupling models? Given that the main contribution (as I see it) is the coupler yet this is not novel i think the authors have the challenge to clearly articulate

what their contribution is. I encourage them to revise their manuscript accordingly to do this.

1. Introduction. The background provided in the introduction is a nice overview.

2. Lines 91-97, the authors should also include TerrSysMP system (Shrestha et al MWR 2104) in this list and perhaps the numerous follow up studies using this platform. The platform is particularly important as it couples the same LSM as used in this study (CLM 3.5, now 4.5) coupled to an integrated hydrology model. As examples, the authors of this platform have used it for fully-coupled studies over all of Europe (Keune et al JGR-A 2016) and for high resolution simulation (Gebler et al JoH 2017). I strikes me that these studies are much more advanced than the current effort and should be used to demonstrate how the current study is advancing the science.

3. Lines 103-104, the sentence is confusing. Do you mean that sometimes models agree and sometimes they don't?

4. Paragraph starting at line 107. This paragraph should be re-structed. One of the main criticisms I have of this work is the lack of novelty. This paragraph is one of the main places the authors can distinguish their work from prior studies. They don't in fact show scalability of either code and the other two points are somewhat weak science goals. I think restructuring this paragraph will help the authors develop a manuscript that is better organized and articulates the contributions made by this work.

5. Integrated hydrology models are such (and not just Richards' solvers) because they solve a form of the shallow water equations and Richards' equation in a globally implicit manner. It is unclear that PFLOTRAN has a surface component, so is it an integrated code?

6. Lines 205-220. As I see it, the coupler is the only potential contribution made in this work. The description needs to be much more detailed. What fluxes and states are passed between the two codes? How is the gridding handled? How is the parallelization accomplished for tiling in CLM and cells in PFLOTRAN? How is the grid overlap between soil column and 3D mesh approached, is the 3D Richards' formulation used in place of CLM or is there some other point where the codes couple? What time integration strategy is used? These are all critical points that should be addressed.

7. Lines 218-220. Surely the authors don't mean this is the first study to couple 3D Richards' equation to a land surface model, that has been documented in the literature for more than a decade. Do the authors mean the PFLOTRAN CLM 4.5 coupling? That isn't novel. This sentence makes the authors sound either disingenuous or naive, either way I think it should be restructured or removed.

8. Verification. There is no section describing the verification of the modeling approach. Prior studies have carefully calculated the energy and water balance of the individual and coupled systems to ensure that nothing in the original formulations has been altered by the coupling and that the coupled system balances water and energy between models. This is a critical missing aspect of the work. It's important to distinguish this from model validation, where a system that is poorly constructed could still be tuned to match observations.

9. PFCLM. The abbreviation PFCLM has been used widely in the literature for the coupled codes ParFlow and CLM. The use here is confusing and a different acronym should be chosen. Also, given the order of calling (PFLOTRAN is a subroutine of CLM 4.5) it seems the CLM component leads, not the hydrology one.

10. Scale. The Hanford test case appears to be at very fine spatial resolution (2m) which violates most of the assumptions made for land-energy fluxes in CLM. The M-O stability and ET formulations use a single-column approach which would almost assuredly break down at this resolution. Studies that do consider this type of fine scale usually use LES formulations for the atmosphere to relax this assumption. The authors need to discuss this and perhaps discard the 2m case.

---

## Editor Comment (EC1) · J. Kala (Editor) · 3 May 2017

This manuscript is well written. It describes the evaluation and coupling of CLM4.5, a widely used LSM, to PFLOTRAN, a subsurface model. The individual codes and model coupling are well described. The simulations are evaluated under real-life conditions. The paper fits the scope of GMD very well. The paper can be accepted following the following revisions:

The most major revision required to this paper is that results with CLM4.5 alone, without coupling to PFLOTRAN, are not presented. The reader hence does not get an idea of

the added benefit of running the LSM coupled to a sophisticated subsurface model. What are the differences in the surface heat fluxes by running CLM alone versus the coupled system? If sub-surface flows have an influence on the surface energy balance, then it needs to proved that it is actually worth the effort to run the coupled system?

The abstract should mention the 3 different spatial resolutions used, especially as it is stated later that spatial resolution had a significant impact.

In the abstract, it is also stated that including lateral subsurface flow impacted (I suggest using the word influenced rather than impacted) the surface energy budget and subsurface transport. How?

At the end of the abstract, it is stated that this coupled system could be used to study land-atmosphere interactions. This is not really correct as this current modeling system does not include a dynamic atmospheric component? You ran the model with prescribed meteorology. You cannot really make this conclusion.

Line 67 – The acronym ESM does not seem to be have previously defined? Is this acronym really necessary?

The introduction gives no indication why coupling CLM and PFLOTRAN is a good and worthwhile idea. Why these two models? If CLM has been coupled to other subsurface models such as PAWS, then what makes PFLOTRAN more advantageous than PAWS? While I have no doubt coupling CLM and PFLOTRAN is a great idea, you need to explain a bit more on why this is the case. Provide a bit more background, one paragraph should do.

The paper tends to make use of many acronyms, and many of these do not seem necessary. Please only use acronyms where it is warranted. For example, the LEAF acronym is only used once, so there is no point in defining it if you don't use it again. Please carefully review all your acronyms.

In Figure 1, some of the arrows do not seem to make sense to me. CLM links directly to

PFLOTRAN Initialize, execute and finalize. Surely, CLM should only link to PLFOTRAN initialize, when then links to PFLOTRAN execute, then finalize. Also, according to your diagram, no information flows back from PFLOTRAN to CLM? Your diagram suggests that there is no two-way coupling? But the text state that soil moisture and hydraulic properties from PFLOTRAN and given back to CLM. Your flowchart does not really show this?

Use m day-1 rather than m d-1.

Figure 4 – Sorry I can hardly read any of the figure titles, please make these larger and more easily readable.

Lines 359 – 361: You state that cold month were excluded from the analysis as you end up with division by zero issues when LH becomes close to zero. That's why most people use the evaporative fraction (EF), rather than the Bowen ratio. With EF, you take the ratio of latent to the sum of sensible and latent, hence, you will not have division by zero issues. You should use EF rather than Bowen ratio.

Section 4.1 – Please use model evaluation rather than model validation. Validation implies the model is already correct to start with and you are therefore validating it. This is of course never true of any model.

Line 418: Looking at Figure 7(a) and 8, I find it hard to get an accurate idea of the differences, could you please plot the difference instead?

Figure 10 – can you please remove the textbox at the bottom (CONTOUR FROM . . . . . . .). Looks like an NCL plot to me. . . . I'm sure you can remove this: Âăres@cnInfoLabelOn = False

Line 447: Don't start a sentence with And.

Your use of the 2 m simulation as a surrogate truth is fine, given a lack of observations of what is being simulated. However, you cannot really say simulation x outperformed simulation y (line 482), explain why one simulation appears more realistic, but I am not

comfortable with the word "outperform".

It would have been really interesting if you ran your model over a site for which observationally derived flux tower estimates of H and LE are available, such that you could then assess if this coupled system actually improves on CLM4.5 alone in simulating surface energy fluxes. I do understand that locations where Flux tower data are currently available (e.g., the FLUXNET network), may not necessarily be regions where the hydrology is interesting enough to warrant the use of such a model. You do however, need to acknowledge somewhere that the model needs to be evaluated against actual observations of surface fluxes.

Code availability: We had a recent discussion among GMD editors, and the point of the Code Availability section is to ensure the reproducibility. What we want is the exact code used for this paper. It is of course understandable that the code is still under development, however, we request you make the version of the code used for this paper available. If this is already on bitbucket or github, it is quite easy to make the revision/branch used for this study on ZENODO, which is the preferred repository for code as per GMD guidelines as it will generate an actual DOI for the code:

http://www.geoscientific-model-development.net/about/code_and_data_policy.html

If you do a quick search on ZENODO, you will find several codes which point to github/bitbucket repositories, but a "frozen" version of the code used can be directly obtained from ZENODO, rather than a user having to work out which branch/revision of your code was used in the paper from the github/bitbucket repo.

---

## Author Comment (AC2) · 10 Jul 2017

**Coupling a three-dimensional subsurface flow and transport model with a land surface model to simulate stream-aquifer-land interactions (CPv1.0) [MS No.: gmd-2017-35]**

**Responses to review comments**

**J. Kala (Editor) j.kala@murdoch.edu.au**

*This manuscript is well written. It describes the evaluation and coupling of CLM4.5, a widely used LSM, to PFLOTRAN, a subsurface model. The individual codes and model coupling are well described. The simulations are evaluated under real-life conditions. The paper fits the scope of GMD very well. The paper can be accepted following the following revisions:*

*The most major revision required to this paper is that results with CLM4.5 alone, without coupling to PFLOTRAN, are not presented. The reader hence does not get an idea of the added benefit of running the LSM coupled to a sophisticated subsurface model.*

**Response**: We have added a CLM4.5 standalone simulation for comparison. Please see our response to the referee #1 for more details.

*What are the differences in the surface heat fluxes by running CLM alone versus the coupled system? If sub-surface flows have an influence on the surface energy balance, then it needs to proved that it is actually worth the effort to run the coupled system?*

**Response**: Please see the response to the question from Referee #1 above. The difference of subsurface flows on surface energy balance is significant between figure S4 (CLM4.5) and Figure 8(a) (i.e., CPv1) due to reasons stated in the response to Referee #1.

*The abstract should mention the 3 different spatial resolutions used, especially as it is stated later that spatial resolution had a significant impact. In the abstract, it is also stated that including lateral subsurface flow impacted (I suggest using the word influenced rather than impacted) the surface energy budget and subsurface transport. How?*

**Response**: Thanks for the constructive suggestions. We have added the spatial resolutions to the abstract, changed "impacted" to "influenced", and added a phrase to discuss the reason why lateral subsurface flow could impact surface energy budget and subsurface transport. Please check the revised abstract for details.

*At the end of the abstract, it is stated that this coupled system could be used to study land-atmosphere interactions. This is not really correct as this current modeling system does not*

*include a dynamic atmospheric component? You ran the model with prescribed meteorology. You cannot really make this conclusion.*

**Response**: Thanks. We have removed that sentence in the revised abstract.

*Line 67 – The acronym ESM does not seem to be have previously defined? Is this acronym really necessary?*

**Response**: Thanks. We have deleted the acronym ESM.

*The introduction gives no indication why coupling CLM and PFLOTRAN is a good and worthwhile idea. Why these two models? If CLM has been coupled to other subsurface models such as PAWS, then what makes PFLOTRAN more advantageous than PAWS? While I have no doubt coupling CLM and PFLOTRAN is a great idea, you need to explain a bit more on why this is the case. Provide a bit more background, one paragraph should do.*

**Response**:  In response to this comment and that from the Referee #2, we have added literature reviews and discussions to elaborate the scientific motivation of this study (i.e., the potential of exploring the fully coupled aquifer-soil-vegetation-atmosphere continuum using an integrated model) in Section 1 of the revised manuscript. We also added discussions on how CP v1 differs from CLM-PAWS section 1 of the revised manuscript as follows:

"The developments of the integrated models have enabled scientific explorations of interactions and feedback mechanisms in the aquifer-soil-vegetation-atmosphere continuum using a holistic and physically based approach (Shrestha et al., 2014;Gilbert et al., 2017). Compared to simulations of regional climate models coupled to traditional LSMs, such a physically based approach shows less sensitivity to uncertainty in the subsurface hydraulic characteristics that could propagate from deep subsurface to free troposphere (Keune et al., 2016), while other physical representations (e.g., parameterizations in evaporation and transpiration, atmospheric boundary layer schemes) could have significant effects on the simulations as well (Sulis et al., 2017). Therefore, it is of great scientific interest to further develop the integrated models and benchmarks to achieve improved understanding of complex interactions in the fully coupled Earth system.

Motivated by the great potentials of using an integrated model to explore Earth system dynamics, the objective of this study is three-fold. First, we aim to document the development of a coupled land surface and subsurface model as a first step toward a new integrated model, featuring the two-way coupling between two highly-scalable and state-of-the-art open-source codes: CLM4.5 [*Oleson et al.*, 2013] and a reactive transport model PFLOTRAN [*Lichtner et al.*, 2015]. The coupled model mechanistically represents the two-way exchange of water and solute mass between aquifers and river, as well as land-atmosphere exchange of water and energy. The coupled model is therefore named as CP v1.0 hereafter. We note that in recent years, efforts have been made to implement carbon–nitrogen decomposition, nitrification,

denitrification, and plant uptake from CLM4.5 in the form of a reaction network solved by PFLOTRAN to enable the coupling of biogeochemical processes between the two models [*Tang et al.*, 2016]. In addition, although PAWS is coupled to the same version of CLM (i.e., CLM4.5) (Ji et al., 2015;Pau et al., 2016), PFLOTRAN resolves the subsurface in a 3-D fashion, while PAWS approximates the 3D Richards equation by divide the subsurface into an unsaturated domain represented by the 1-D Richards Equation coupled with 3D saturated groundwater flow equation for subsurface flow, by assuming that there is no horizontal flow in unsaturated portion of soil, and that lateral flux in saturated portion is evenly distributed."

*The paper tends to make use of many acronyms, and many of these do not seem necessary. Please only use acronyms where it is warranted. For example, the LEAF acronym is only used once, so there is no point in defining it if you don't use it again. Please carefully review all your acronyms.*

Response: Thanks. We have deleted all unnecessary acronyms.

*In Figure 1, some of the arrows do not seem to make sense to me. CLM links directly to PFLOTRAN Initialize, execute and finalize. Surely, CLM should only link to PLFOTRAN initialize, when then links to PFLOTRAN execute, then finalize. Also, according to your diagram, no information flows back from PFLOTRAN to CLM? Your diagram suggests that there is no two-way coupling? But the text state that soil moisture and hydraulic properties from PFLOTRAN and given back to CLM. Your flowchart does not really show this? Use m day-1 rather than m d-1.*

Response: In response to all reviewers' comment regarding model coupling, we have significantly revised the technical details about model coupling in Section 2.3 (see below) and added a new schematic describing the model coupling (Figure 1). We have updated Figure 1 to better represent the two-way model coupling. We also have changed the unit "m d-1" to "m day-1".

"In this study, CLM4.5's one-dimensional models for flow in unsaturated (Zeng and Decker, 2009) and saturated (Niu et al., 2007) zones are replaced by PFLOTRAN's RICHRADS mode to simulate unsaturated-saturated flow within the three-dimensional subsurface domain. Although PFLOTRAN is also capable of simulating coupled flow and thermal processes in the subsurface including explicit representation of liquid-ice phase (Karra et al., 2014), as well as, soil nutrient cycles, (Hammond and Lichtner, 2010;Zachara et al., 2016;Tang et al., 2016),  those processes are not coupled between the two models in this study. A schematic representation of the coupling between CLM4.5 and PFLOTRAN is shown in Figure 1. A model coupling interface based on PETSc data structures was developed to couple the two models and the interface includes some key design features of the CESM coupler [*Craig et al.*, 2012]. The model coupling interface allows each model grid to have a different spatial resolution and domain decomposition across multiple processors. While CLM4.5 uses a round-robin decomposition approach, PFLOTRAN employs domain decomposition via PETSc (Figure 1a). Interpolation of gridded data from one model onto the grids of the other is done through sparse matrix vector

multiplication. As a preprocessing step, sparse weight matrices for interpolating data between the two models are saved as mapping files. Analogous to the CESM coupler, the mapping files are saved in a format similar to the mapping files produced by the ESMF_RegridWeightGen (https://www.earthsystemcog.org/projects/regridweightgen). ESMF regridding tools provide multiple interpolation methods (conservative, bilinear, and nearest neighbor) to generate the sparse weight matrix. In this work, we have used a conservative remapping method to interpolate data between CLM and PFLOTRAN. During model initialization, the model coupling interface first collectively reads all required sparse matrices. Next, the model coupling interface reassembles local sparse matrices after accounting for domain decomposition of each model (figures 1b and 1c). "

*Figure 4 – Sorry I can hardly read any of the figure titles, please make these larger and more easily readable.*

**Response**: Thanks for pointing this out. We have made the font of the figure titles bigger in the revised manuscript, including those in supplementary materials (figures S1 and S2).

*Lines 359 – 361: You state that cold month were excluded from the analysis as you end up with division by zero issues when LH becomes close to zero. That's why most people use the evaporative fraction (EF), rather than the Bowen ratio. With EF, you take the ratio of latent to the sum of sensible and latent, hence, you will not have division by zero issues. You should use EF rather than Bowen ratio.*

**Response**: Thanks for the great suggestion. We have redone the analysis using EF instead of Bowen ratio, and modified the figure and text in the revised manuscript correspondingly.

*Section 4.1 – Please use model evaluation rather than model validation. Validation implies the model is already correct to start with and you are therefore validating it. This is of course never true of any model.*

**Response:** Thanks for the great suggestion. We have changed the section title to be "Model evaluation".

*Line 418: Looking at Figure 7(a) and 8, I find it hard to get an accurate idea of the differences, could you please plot the difference instead? Figure 10 – can you please remove the textbox at the bottom (CONTOUR FROM.......). Looks like an NCL plot to me*
*I'm sure you can remove this: ares@cnInfoLabelOn = False*

**Response**:  We have modified the figures as suggested. Please check the revised manuscript for details. We also modified the figures in the supplementary material accordingly to be consistent.

*Line 447: Don't start a sentence with And. Your use of the 2 m simulation as a surrogate truth is fine, given a lack of observations of what is being simulated. However, you cannot really say simulation x outperformed simulation y (line 482), explain why one simulation appears more realistic, but I am not comfortable with the word "outperform". It would have been really interesting if you ran your model over a site for which observationally derived flux tower estimates of H and LE are available, such that you could then assess if this coupled system actually improves on CLM4.5 alone in simulating surface energy fluxes. I do understand that locations where Flux tower data are currently available (e.g., the FLUXNET network), may not necessarily be regions where the hydrology is interesting enough to warrant the use of such a model. You do however, need to acknowledge somewhere that the model needs to be evaluated against actual observations of surface fluxes.*

**Response**: Thanks for the great suggestions. We have modified the sentences as suggested in the revised manuscript. In fact, two flux towers have been installed along the Hanford reach for this purpose but the analysis of the flux measurement is still preliminary. In addition, both towers are a little distant from the modeling domain to satisfy the requirements of eddy covariance measurements. Nevertheless, we also added discussions on the need of evaluating the model using eddy covariance measurements in section 5 of the revised manuscript.

*Code availability: We had a recent discussion among GMD editors, and the point of the Code Availability section is to ensure the reproducibility. What we want is the exact code used for this paper. It is of course understandable that the code is still under development, however, we request you make the version of the code used for this paper available. If this is already on bitbucket or github, it is quite easy to make the revision/branch used for this study on ZENODO, which is the preferred repository for code as per GMD guidelines as it will generate an actual DOI for the code: http://www.geoscientific-model-development.net/about/code_and_data_policy.html*
*If you do a quick search on ZENODO, you will find several codes which point to github/bitbucket repositories, but a "frozen" version of the code used can be directly obtained from ZENODO, rather than a user having to work out which branch/revision of your code was used in the paper from the github/bitbucket repo*

**Response**: Thanks for the instruction. The model and data have been made

publicly available at

- [https://bitbucket.org/clm_pflotran/clm-pflotran-trunk](https://bitbucket.org/clm_pflotran/clm-pflotran-trunk): CLM code
- [https://bitbucket.org/clm_pflotran/pflotran-clm-trunk](https://bitbucket.org/clm_pflotran/pflotran-clm-trunk): PFLOTRAN code
- [https://bitbucket.org/pnnl_sbr_sfa/notes-for-gmd-2017-35](https://bitbucket.org/pnnl_sbr_sfa/notes-for-gmd-2017-35): Data

The README file in the notes-for-gmd-2017-35 repository provides detailed notes on how to create, compile, and run a simulation. Once the manuscript is accepted, we will start porting the frozen version of the code to ZENODO.

**Reference:**

Gilbert, J. M., Maxwell, R. M., and Gochis, D. J.: Effects of Water-Table Configuration on the Planetary Boundary Layer over the San Joaquin River Watershed, California, Journal of Hydrometeorology, 18, 1471-1488, 10.1175/jhm-d-16-0134.1, 2017.

Hammond, G. E., and Lichtner, P. C.: Field-scale model for the natural attenuation of uranium at the Hanford 300 Area using high-performance computing, Water Resources Research, 46, n/a-n/a, 10.1029/2009WR008819, 2010.

Ji, X., Shen, C., and Riley, W. J.: Temporal evolution of soil moisture statistical fractal and controls by soil texture and regional groundwater flow, Advances in Water Resources, 86, Part A, 155-169, http://dx.doi.org/10.1016/j.advwatres.2015.09.027, 2015.

Karra, S., Painter, S. L., and Lichtner, P. C.: Three-phase numerical model for subsurface hydrology in permafrost-affected regions (PFLOTRAN-ICE v1.0), The Cryosphere, 8, 1935-1950, 10.5194/tc-8-1935-2014, 2014.

Keune, J., Gasper, F., Goergen, K., Hense, A., Shrestha, P., Sulis, M., and Kollet, S.: Studying the influence of groundwater representations on land surface-atmosphere feedbacks during the European heat wave in 2003, Journal of Geophysical Research: Atmospheres, 121, 13,301-313,325, 10.1002/2016JD025426, 2016.

Niu, G.-Y., Yang, Z.-L., Dickinson, R. E., Gulden, L. E., and Su, H.: Development of a simple groundwater model for use in climate models and evaluation with Gravity Recovery and Climate Experiment data, Journal of Geophysical Research: Atmospheres, 112, n/a-n/a, 10.1029/2006JD007522, 2007.

Pau, G. S. H., Shen, C., Riley, W. J., and Liu, Y.: Accurate and efficient prediction of fine-resolution hydrologic and carbon dynamic simulations from coarse-resolution models, Water Resources Research, 52, 791-812, 10.1002/2015WR017782, 2016.

Shrestha, P., Sulis, M., Masbou, M., Kollet, S., and Simmer, C.: A Scale-Consistent Terrestrial Systems Modeling Platform Based on COSMO, CLM, and ParFlow, Monthly Weather Review, 142, 3466-3483, 10.1175/mwr-d-14-00029.1, 2014.

Sulis, M., Williams, J. L., Shrestha, P., Diederich, M., Simmer, C., Kollet, S. J., and Maxwell, R. M.: Coupling Groundwater, Vegetation, and Atmospheric Processes: A Comparison of Two Integrated Models, Journal of Hydrometeorology, 18, 1489-1511, 10.1175/jhm-d-16-0159.1, 2017.

Tang, G., Yuan, F., Bisht, G., Hammond, G. E., Lichtner, P. C., Kumar, J., Mills, R. T., Xu, X., Andre, B., Hoffman, F. M., Painter, S. L., and Thornton, P. E.: Addressing numerical challenges in introducing a reactive transport code into a land surface model: a biogeochemical modeling proof-of-concept with CLM–PFLOTRAN 1.0, Geosci. Model Dev., 9, 927-946, 10.5194/gmd-9-927-2016, 2016.

Zachara, J. M., Chen, X., Murray, C., and Hammond, G.: River stage influences on uranium transport in a hydrologically dynamic groundwater-surface water transition zone, Water Resources Research, 52, 1568-1590, 10.1002/2015WR018009, 2016.

Zeng, X., and Decker, M.: Improving the Numerical Solution of Soil Moisture–Based Richards Equation for Land Models with a Deep or Shallow Water Table, Journal of Hydrometeorology, 10, 308-319, 10.1175/2008JHM1011.1, 2009.

---

## Author Comment (AC3) · 10 Jul 2017

**Coupling a three-dimensional subsurface flow and transport model with a land surface model to simulate stream-aquifer-land interactions (CPv1.0) [MS No.: gmd-2017-35]**

**Responses to review comments**

**Anonymous Referee #2**

*In this manuscript the authors document development and application of a coupled Richards' equation solver (PFLOTRAN) with a land surface model (CLM 4.5) and apply it to a test problem developed from an intensely observed floodplain system. This manuscript is generally clearly written but in my opinion needs to better articulate its contributions given the prior work on this topic. I have specific comments below that need to be addressed before the suitability of this work for GMD can be assessed.*

*The larger comments are ones of contribution, what does this work want to contribute to our understanding of coupling models? Given that the main contribution (as I see it) is the coupler yet this is not novel i think the authors have the challenge to clearly articulate what their contribution is. I encourage them to revise their manuscript accordingly to do this.*

1*. Introduction. The background provided in the introduction is a nice overview.*

**Response**: Thanks for the positive comments.

2. *Lines 91-97, the authors should also include TerrSysMP system (Shrestha et al MWR 2104) in this list and perhaps the numerous follow up studies using this platform. The platform is particularly important as it couples the same LSM as used in this study (CLM 3.5, now 4.5) coupled to an integrated hydrology model. As examples, the authors of this platform have used it for fully-coupled studies over all of Europe (Keune et al. JGR-A 2016) and for high resolution simulation (Gebler et al JoH 2017). It strikes me that these studies are much more advanced than the current effort and should be used to demonstrate how the current study is advancing the science.*

**Response**: Thanks for the great suggestion. We have added reviews on these studies in the introduction section of the revised manuscripts, and added discussions on how such coupled models could advance science in section 1 of the revised manuscript as follows:

  "The developments of the integrated models have enabled scientific explorations of interactions and feedback mechanisms in the aquifer-soil-vegetation-atmosphere continuum using a holistic and physically based approach (Shrestha et al., 2014;Gilbert et al., 2017). Compared to simulations of regional climate models coupled to traditional LSMs, such a physically based approach shows less sensitivity to uncertainty in the subsurface hydraulic

characteristics that could propagate from deep subsurface to free troposphere (Keune et al., 2016), while other physical representations (e.g., parameterizations in evaporation and transpiration, atmospheric boundary layer schemes) could have significant effects on the simulations as well (Sulis et al., 2017). Therefore, it is of great scientific interest to further develop the integrated models and benchmarks to achieve improved understanding of complex interactions in the fully coupled Earth system."

3. *Lines 103-104, the sentence is confusing. Do you mean that sometimes models agree and sometimes they don't?*

**Response**: Thanks for pointing this out. We have modified this sentence as follows in the revised manuscript:

"However, as a result of difference in physical process representations and numerical solution approaches in terms of (1) the coupling between the variably saturated groundwater and surface water flow; (2) representation of surface water flow; and (3) implementation of subsurface heterogeneity in the existing integrated models, significant discrepancies exist in their results when the models were applied to highly nonlinear problems with heterogeneity and complex water table dynamics, while many of the models show good agreement for simpler test cases where traditional runoff generation mechanisms (i.e., saturation and infiltration excess runoff)  apply (Kollet et al., 2017;Maxwell et al., 2014)."

4. *Paragraph starting at line 107. This paragraph should be re-structured. One of the main criticisms I have of this work is the lack of novelty. This paragraph is one of the main places the authors can distinguish their work from prior studies. They don't in fact show scalability of either code and the other two points are somewhat weak science goals. I think restructuring this paragraph will help the authors develop a manuscript that is better organized and articulates the contributions made by this work.*

**Response**: Thanks for the constructive comments. We have revised this section significantly to include discussions on the scientific potential of integrated models based on recent studies. We also revised the coupling section to provide more details on how the coupling was achieved. Please check the revised manuscript to see if the revisions are satisfactory.

5. *Integrated hydrology models are such (and not just Richards' solvers) because they solve a form of the shallow water equations and Richards' equation in a globally implicit manner. It is unclear that PFLOTRAN has a surface component, so is it an integrated code?*

**Response**: In this study, a prognostic model for simulating river stage dynamics is not included. Instead, a river stage boundary condition was applied to PFLOTRAN to capture observed and hypothetical river stage scenarios. Even though a shallow water equation implementation in PFLOTRAN is under testing, it is premature and warrants a standalone study to assess its performance. Therefore, we have modified the text throughout the revised manuscript to remove ambiguity in this regard. We also added the need of implementing a surface flow component to

qualify the model as an integrated hydrology model into the discussion section. Please check the revised manuscript for detail.

6. *Lines 205-220. As I see it, the coupler is the only potential contribution made in this work. The description needs to be much more detailed. What fluxes and states are passed between the two codes? How is the gridding handled? How is the parallelization accomplished for tiling in CLM and cells in PFLOTRAN? How is the grid overlap between soil column and 3D mesh approached, is the 3D Richards' formulation used in place of CLM or is there some other point where the codes couple? What time integration strategy is used? These are all critical points that should be addressed.*

**Response**: In response to all reviewers' comment regarding model coupling, we have significantly revised the technical details about model coupling in Section 2.3 and added a new schematic (Figure 1) to describe the two-way model coupling. We attempted to answer all the specific questions of this comment. The models are coupled two-ways and we have updated Figure 1 to better represent model coupling as follows. Please section 2.3 of the revised manuscript for detailed description on this coupling.

[Figure]

Figure R1. Schematic representations of the model coupling interface of CP v1.0. (a) Domain decomposition of a hypothetical CLM and PFLOTRAN domain comprising of 4x1x7 and 4x1x5 grids in x, y, and z directions across two processors as shown in blue and green. (b) Mapping of water fluxes from CLM onto PFLOTRAN domain via a local sparse matrix vector product for grids on processor 1. (c) Mapping of updated soil moisture from PFLOTRAN onto CLM domain via a local sparse matrix vector product for grids on processor 1.

*7. Lines 218-220. Surely the authors don't mean this is the first study to couple 3D Richards' equation to a land surface model, that has been documented in the literature for more than a decade. Do the authors mean the PFLOTRAN CLM 4.5 coupling? That isn't novel. This*

*sentence makes the authors sound either disingenuous or naive, either way I think it should be restructured or removed.*

**Response**: Thanks for the suggestion. We have deleted this sentence.

*8. Verification. There is no section describing the verification of the modeling approach. Prior studies have carefully calculated the energy and water balance of the individual and coupled systems to ensure that nothing in the original formulations has been altered by the coupling and that the coupled system balances water and energy between models. This is a critical missing aspect of the work. It's important to distinguish this from model validation, where a system that is poorly constructed could still be tuned to match observations.*

**Response:** We agree that a thorough evaluation of the energy and water balance terms is needed for the system. To address the reviewer's concern, we have (1) verified that the subsurface solver by evaluating the mass balance errors for each time step and added the figure to the supplementary material; and (2) verified that the surface energy balance and added the figure to the supplementary material. Discussions on these figures were added in section 3.1 of the revised manuscript.

*9. PFCLM. The abbreviation PFCLM has been used widely in the literature for the coupled codes ParFlow and CLM. The use here is confusing and a different acronym should be chosen. Also, given the order of calling (PFLOTRAN is a subroutine of CLM 4.5) it seems the CLM component leads, not the hydrology one.*

**Response**: Thanks for the constructive suggestion. We have changed the model name to be CP v1.0 to be consistent with the sequence of coupling, and to differentiate the model from ParFlow-CLM and previous coupled versions of CLM4.5 and PFLOTRAN. We have modified all occurrences of the names in the text and figures. Please check the revised manuscripts for details.

*10. Scale. The Hanford test case appears to be at very fine spatial resolution (2m) which violates most of the assumptions made for land-energy fluxes in CLM. The M-O stability and ET formulations use a single-column approach which would almost assuredly break down at this resolution. Studies that do consider this type of fine scale usually use LES formulations for the atmosphere to relax this assumption. The authors need to discuss this and perhaps discard the 2m case.*

**Response:** As noted by the reviewer, CLM uses the M-O similarity theory to compute friction velocity and other exchange coefficients that provide the basis to estimate surface heat and moisture fluxes. It is also common knowledge that the M-O similarity theory is only valid when the surface layer depth $z \gg z0$, where $z0$ is the aerodynamic roughness length. As reviewed in Basu and Lacser (2017), It is highly recommended that $z > 50z_0$ to ensure that the lower atmospheric level is higher than the size of surface wakes in the roughness sublayer, which

should be proportional to horizontal grid spacing to guarantee the validity of the M-O similarity theory (Arnqvist and Bergström, 2015).

In our simulations, the majority of the Hanford 300A domain is covered by bare soil ($z_0$ = 0.01 m), grass ($z_0$ = 0.013 m), shrubs ($z_0$ = 0.026-0.043 m), and riparian trees (varies across the seasons, $z_0$ = 0.008 m when LAI = 2 in the summer and $z_0$ = 1.4 when LAI = 0 in the winter). Therefore, under most condition a 2-m resolution is sufficiently coarse to ensure the validity of the M-O similarly theory, except for the grid cells covered by riparian trees in the winter. However, in our simulations, the wintertime latent heat and sensible heat fluxes approach zero due to extremely low energy inputs in the winter. In addition, the 2-m resolution simulations are valuable for verifying subsurface simulations. Therefore, after careful considerations, we decide to keep the 2-m simulations, but added discussion on potential issues when the model is run at such a resolution section 5 in the revised manuscript. We hope such a treatment could alleviate problems associated with the scale of model applicability.

**Reference:**

Arnqvist, J., and Bergström, H.: Flux-profile relation with roughness sublayer correction, Quarterly Journal of the Royal Meteorological Society, 141, 1191-1197, 10.1002/qj.2426, 2015.

Basu, S., and Lacser, A.: A Cautionary Note on the Use of Monin–Obukhov Similarity Theory in Very High-Resolution Large-Eddy Simulations, Boundary-Layer Meteorology, 163, 351-355, 10.1007/s10546-016-0225-y, 2017.

Gilbert, J. M., Maxwell, R. M., and Gochis, D. J.: Effects of Water-Table Configuration on the Planetary Boundary Layer over the San Joaquin River Watershed, California, Journal of Hydrometeorology, 18, 1471-1488, 10.1175/jhm-d-16-0134.1, 2017.

Keune, J., Gasper, F., Goergen, K., Hense, A., Shrestha, P., Sulis, M., and Kollet, S.: Studying the influence of groundwater representations on land surface-atmosphere feedbacks during the European heat wave in 2003, Journal of Geophysical Research: Atmospheres, 121, 13,301-313,325, 10.1002/2016JD025426, 2016.

Kollet, S., Sulis, M., Maxwell, R. M., Paniconi, C., Putti, M., Bertoldi, G., Coon, E. T., Cordano, E., Endrizzi, S., Kikinzon, E., Mouche, E., Mügler, C., Park, Y.-J., Refsgaard, J. C., Stisen, S., and Sudicky, E.: The integrated hydrologic model intercomparison project, IH-MIP2: A second set of benchmark results to diagnose integrated hydrology and feedbacks, Water Resources Research, 53, 867-890, 10.1002/2016WR019191, 2017.

Maxwell, R. M., Putti, M., Meyerhoff, S., Delfs, J.-O., Ferguson, I. M., Ivanov, V., Kim, J., Kolditz, O., Kollet, S. J., Kumar, M., Lopez, S., Niu, J., Paniconi, C., Park, Y.-J., Phanikumar, M. S., Shen, C., Sudicky, E. A., and Sulis, M.: Surface-subsurface model intercomparison: A first set of benchmark results to diagnose integrated hydrology and feedbacks, Water Resources Research, 50, 1531-1549, 10.1002/2013WR013725, 2014.

Shrestha, P., Sulis, M., Masbou, M., Kollet, S., and Simmer, C.: A Scale-Consistent Terrestrial Systems Modeling Platform Based on COSMO, CLM, and ParFlow, Monthly Weather Review, 142, 3466-3483, 10.1175/mwr-d-14-00029.1, 2014.

Sulis, M., Williams, J. L., Shrestha, P., Diederich, M., Simmer, C., Kollet, S. J., and Maxwell, R. M.: Coupling Groundwater, Vegetation, and Atmospheric Processes: A Comparison of Two Integrated Models, Journal of Hydrometeorology, 18, 1489-1511, 10.1175/jhm-d-16-0159.1, 2017.

---

## Author Response (AR1)

Proudly Operated by Battelle Since 1965

902 Battelle Boulevard P.O. Box 999, MSIN XX-XX Richland, WA 99352 (509) 375-6827 Maoyi.Huang@pnnl.gov

www.pnnl.gov

10 July 2017

Dear Editor:

Please find enclosed for a revised manuscript entitled "*Coupling a three-dimensional subsurface flow and transport model with a land surface model to simulate stream-aquifer-land interactions (CP v1.0)*" which I am submitting for consideration of publication as a model description paper in Geoscientific Model Development.

Significant modifications have been made in the revised version compared to the original submission and are summarized as follows:

- 1. The model name has been changed from PFLOTRAN\_CLM v1.0 to CP v1.0 to address the concern from reviewers on the coupling sequence and potential overlap with existing models;
- 2. The model coupling section (section 2.3) has been thoroughly revised to answer questions from the reviewers;
- 3. The introduction section (section 1) has been revised to include recent developments in the field and to provide stronger scientific motivations for developing a coupled model;
- 4. Detailed discussions on limitations of CP v1.0 are added to section 5 of the revised manuscript.
- 5. Additional figures are provided, and the original figures are revised in response to reviewers' comments.
- 6. The code availability section has been revised to include publicly accessible repositories for the model codes, tutorial, driving scripts, and datasets used in the paper.

We sincerely hope that the revisions could make the manuscript suitable for publication in Geoscientific Model Development. Thanks for your consideration and we look forward to hearing from you.

Sincerely,

Maoyi Huang

Maoyi Huang, Ph.D. Staff Scientist Atmospheric Sciences and Global Change Division Pacific Northwest National Laboratory

**Coupling a three-dimensional subsurface flow and transport model with a land surface model to simulate stream-aquifer-land interactions (CPv1.0) [MS No.: gmd-2017-35]**

**Responses to review comments**

**Anonymous Referee #1:**

The authors present a new coupled version of CLM4.5 and PFLOTRAN, and demonstrate the impacts of resolution, horizontal fluxes, and river stage height in simulating groundwater levels and turbulent fluxes between the land and the atmosphere.

The authors demonstrate that the new model is capable of simulating the observed water table depth, independent of the model resolution. The authors show PF-CLM results when there is no lateral subsurface exchange. Does this produce the exact same results as CLM without PFLOTRAN? If not CLM should be included in the manuscript. If so the authors should state that running PF-CLM without horizontal transfer gives identical results to CLM.

**Response:**

Thanks for the suggestion. We understand that the standalone CLM4.5 could serve as a good reference for most readers. Therefore, we have included a figure from the CLM4.5 simulation in the supplementary material (i.e., Figure S4) and added discussions on differences in lines 415-427 of the revised manuscript for clarity.

For information, the reasons that a CLM4.5 standalone was not included in the original manuscript were:

(1) The subsurface domain in CLM4.5 for hydrologic processes only extends to 3.8 m below the surface, while in CPv1 subsurface hydrologic processes are simulated ~30m below the surface;

(2) As reviewed in section 2.2 of the original manuscript, CLM4.5 uses TOPMODEL-based parameterizations to simulate surface and subsurface runoff, as well as mean groundwater table depth using formulations derived from catchment hydrology that do not apply at the field site of interest;

(3) The key hydrologic progresses (i.e., the exchange of river water and groundwater at the east boundary and lateral transfer of water at all other boundaries) that affect the hydrologic budget of the system are missing from CLM4.5.

The description of the technical details of the coupling needs more explanation. It is cleat that only the soil moisture and hydraulic properties as passed between CLM and PFLOTRAN. However how does this work given that the vertical discretization of CLM differs from PFLOTRAN? The vertical resolution of the subsurface (PFLOTRAN) component is only 0.5

meters, while CLM uses layers from mm to m. How does this impact transpiration? Is the default rooting depth used in CLM? How are the 0.5 meter thick layers mapped to the much thinner layers? Does CLM compute freezing and thawing? Which processes are no longer used by CLM in the coupled version?

**Response:**

In response to all reviewers' comment regarding model coupling, we have significantly revised the technical details about model coupling in Section 2.3 and schematic describing the model coupling in Figure 1 of the revised manuscript.

While the updated section addresses all of the questions raised by the reviewer, we summarize the answers here. The model coupling interface is able to accommodate different vertical and horizontal resolution between the two models. In our present work, both models had the same horizontal resolution, but different vertical resolution. CLM used the default, exponentially varying vertical discretization, but PFLOTRAN had a uniform 0.5 [m] vertical spacing. The vertical extent of the domain in PFLOTRAN is deeper than the CLM domain. The model coupling interface uses conservative interpolation scheme to remap data between two model grids.

Although a study of the changes in computed transpiration due to differences in vertical resolution between CLM and PFLOTRAN is an interesting research investigation, it is beyond the scope of this work. The coupled simulation used the default rooting distribution of CLM. Although PFLOTRAN has a mode that can explicitly handle liquid and ice phase (Karra et al., 2014), in this work, freezing/thaw dynamics was handled by CLM. In this work, CLM's 1D model for flow in unsaturated (Zeng and Decker, 2009) and saturated (Niu et al., 2007) zones are replaced by PFLOTRAN's 3D flow model.

I am having trouble understanding why the grasses away from the river always have near zero latent heat flux (Figure 7a) while the bare ground has a larger latent heat flux? This explains why the latent heat flux only differs over the bare soil surfaces between CLMPF2m and CLMPFv2m. I fail to understand why the bare soil has a higher latent heat flux than the vegetation, especially given that the moisture available to the roots from horizontal transfer should be even greater than the moisture at the surface. The authors need to explain if this is the expected behavior in CLM, or if it is due to the coupling between PFLOTRAN and CLM.

**Response:**

It is a known problem that, in CLM4 and CLM4.5, ET could be enhanced when vegetation is removed. This ET enhancement over bare soil has been documented as a counter-intuitive bias for most unsaturated soils in CLM4 and CLM4.5 simulations (Lawrence et al., 2012;Tang and Riley, 2013a). Tang and Riley (2013a) explored a few potential causes for this likely bias (e.g., soil resistance, litter layer resistance, and numerical time step). They found the implementation of a physically based soil resistance lowered the bias slightly, but concluded that the bias remained (Tang and Riley, 2013b). Meanwhile, in studying ET over semiarid regions, Swenson and Lawrence (2014) proposed another soil resistance formulation to fix this excessive soil

evaporation problem within CLM4.5. While their modification improved the simulated terrestrial water storage anomaly and ET when compared to GRACE data and FLUXNET-MTE data, respectively, the empirical nature of the soil resistance proposed could have underestimated the soil resistance variability when compared to other estimates (Tang and Riley, 2013b).Therefore, this is expected behavior in CLM rather than being introduced by the coupling between CLM and PFLOTRAN. We have added discussions in lines628-642 the revised manuscript.

**Figure 6 should be shown as the difference between the observations and the simulations. This will show much more information concerning how the simulations differ.**

**Response:** We have made changes as suggested. Please check Figure 7 in the revised manuscript for details. We also moved the original figure to the supplementary material as a reference for the readers (i.e., Figure S5).

**References**

Karra, S., Painter, S. L., and Lichtner, P. C.: Three-phase numerical model for subsurface hydrology in permafrost-affected regions (PFLOTRAN-ICE v1.0), The Cryosphere, 8, 1935-1950, 10.5194/tc-8-1935-2014, 2014.

Lawrence, P. J., Feddema, J. J., Bonan, G. B., Meehl, G. A., O'Neill, B. C., Oleson, K. W., Levis, S., Lawrence, D. M., Kluzek, E., Lindsay, K., and Thornton, P. E.: Simulating the Biogeochemical and Biogeophysical Impacts of Transient Land Cover Change and Wood Harvest in the Community Climate System Model (CCSM4) from 1850 to 2100, Journal of Climate, 25, 3071-3095, 10.1175/jcli-d-11-00256.1, 2012.

Niu, G.-Y., Yang, Z.-L., Dickinson, R. E., Gulden, L. E., and Su, H.: Development of a simple groundwater model for use in climate models and evaluation with Gravity Recovery and Climate Experiment data, Journal of Geophysical Research: Atmospheres, 112, n/a-n/a, 10.1029/2006JD007522, 2007.

Swenson, S. C., and Lawrence, D. M.: Assessing a dry surface layer-based soil resistance parameterization for the Community Land Model using GRACE and FLUXNET-MTE data, Journal of Geophysical Research: Atmospheres, 119, 10,299-210,312, 10.1002/2014JD022314, 2014.

Tang, J., and Riley, W. J.: Impacts of a new bare-soil evaporation formulation on site, regional, and global surface energy and water budgets in CLM4, Journal of Advances in Modeling Earth Systems, 5, 558-571, 10.1002/jame.20034, 2013a.

Tang, J. Y., and Riley, W. J.: A new top boundary condition for modeling surface diffusive exchange of a generic volatile tracer: theoretical analysis and application to soil evaporation, Hydrol. Earth Syst. Sci., 17, 873-893, 10.5194/hess-17-873-2013, 2013b.

Zeng, X., and Decker, M.: Improving the Numerical Solution of Soil Moisture–Based Richards Equation for Land Models with a Deep or Shallow Water Table, Journal of Hydrometeorology, 10, 308-319, 10.1175/2008jhm1011.1, 2009.

**Coupling a three-dimensional subsurface flow and transport model with a land surface model to simulate stream-aquifer-land interactions (CPv1.0) [MS No.: gmd-2017-35]**

**Responses to review comments**

**J. Kala (Editor) j.kala@murdoch.edu.au**

This manuscript is well written. It describes the evaluation and coupling of CLM4.5, a widely used LSM, to PFLOTRAN, a subsurface model. The individual codes and model coupling are well described. The simulations are evaluated under real-life conditions. The paper fits the scope of GMD very well. The paper can be accepted following the following revisions:

The most major revision required to this paper is that results with CLM4.5 alone, without coupling to PFLOTRAN, are not presented. The reader hence does not get an idea of the added benefit of running the LSM coupled to a sophisticated subsurface model.

**Response**: We have added a CLM4.5 standalone simulation for comparison. Please see our response to the referee #1 for more details.

What are the differences in the surface heat fluxes by running CLM alone versus the coupled system? If sub-surface flows have an influence on the surface energy balance, then it needs to proved that it is actually worth the effort to run the coupled system?

**Response**: Please see the response to the question from Referee #1 above. The difference of subsurface flows on surface energy balance is significant between figure S4 (CLM4.5) and Figure 8(a) (i.e., CPv1) in the revised manuscript due to reasons stated in the response to Referee #1.

The abstract should mention the 3 different spatial resolutions used, especially as it is stated later that spatial resolution had a significant impact. In the abstract, it is also stated that including lateral subsurface flow impacted (I suggest using the word influenced rather than impacted) the surface energy budget and subsurface transport. How?

**Response:** Thanks for the constructive suggestions. We have added the spatial resolutions to the abstract, changed "impacted" to "influenced", and added a phrase to discuss the reason why lateral subsurface flow could impact surface energy budget and subsurface transport. Please check the revised abstract for details.

At the end of the abstract, it is stated that this coupled system could be used to study landatmosphere interactions. This is not really correct as this current modeling system does not include a dynamic atmospheric component? You ran the model with prescribed meteorology. You cannot really make this conclusion.

**Response:** Thanks. We have removed that sentence in the revised abstract.

Line 67 – The acronym ESM does not seem to be have previously defined? Is this acronym really necessary?

**Response:** Thanks. We have deleted the acronym ESM.

The introduction gives no indication why coupling CLM and PFLOTRAN is a good and worthwhile idea. Why these two models? If CLM has been coupled to other subsurface models such as PAWS, then what makes PFLOTRAN more advantageous than PAWS? While I have no doubt coupling CLM and PFLOTRAN is a great idea, you need to explain a bit more on why this is the case. Provide a bit more background, one paragraph should do.

**Response:** In response to this comment and that from the Referee #2, we have added literature reviews and discussions to elaborate the scientific motivation of this study (i.e., the potential of exploring the fully coupled aquifer-soil-vegetation-atmosphere continuum using an integrated model) in Section 1 of the revised manuscript. We also added discussions on how CP v1 differs from CLM-PAWS in lines 115-142 of the revised manuscript as follows:

"The developments of the integrated models have enabled scientific explorations of interactions and feedback mechanisms in the aquifer-soil-vegetation-atmosphere continuum using a holistic and physically based approach (Shrestha et al., 2014;Gilbert et al., 2017). Compared to simulations of regional climate models coupled to traditional LSMs, such a physically based approach shows less sensitivity to uncertainty in the subsurface hydraulic characteristics that could propagate from deep subsurface to free troposphere (Keune et al., 2016), while other physical representations (e.g., parameterizations in evaporation and transpiration, atmospheric boundary layer schemes) could have significant effects on the simulations as well (Sulis et al., 2017). Therefore, it is of great scientific interest to further develop the integrated models and benchmarks to achieve improved understanding of complex interactions in the fully coupled Earth system.

Motivated by the great potentials of using an integrated model to explore Earth system dynamics, the objective of this study is three-fold. First, we aim to document the development of a coupled land surface and subsurface model as a first step toward a new integrated model, featuring the two-way coupling between two highly-scalable and state-of-the-art open-source codes: CLM4.5 [*Oleson et al.*, 2013] and a reactive transport model PFLOTRAN [*Lichtner et al.*, 2015]. The coupled model mechanistically represents the two-way exchange of water and solute mass between aquifers and river, as well as land-atmosphere exchange of water and

energy. The coupled model is therefore named as CP v1.0 hereafter. We note that in recent years, efforts have been made to implement carbon–nitrogen decomposition, nitrification, denitrification, and plant uptake from CLM4.5 in the form of a reaction network solved by PFLOTRAN to enable the coupling of biogeochemical processes between the two models [*Tang et al.*, 2016]. In addition, although PAWS is coupled to the same version of CLM (i.e., CLM4.5) (Ji et al., 2015;Pau et al., 2016), PFLOTRAN resolves the subsurface in a 3-D fashion, while PAWS approximates the 3D Richards equation by divide the subsurface into an unsaturated domain represented by the 1-D Richards Equation coupled with 3D saturated groundwater flow equation for subsurface flow, by assuming that there is no horizontal flow in unsaturated portion of soil, and that lateral flux in saturated portion is evenly distributed."

The paper tends to make use of many acronyms, and many of these do not seem necessary. Please only use acronyms where it is warranted. For example, the LEAF acronym is only used once, so there is no point in defining it if you don't use it again. Please carefully review all your acronyms.

**Response:** Thanks. We have deleted all unnecessary acronyms.

In Figure 1, some of the arrows do not seem to make sense to me. CLM links directly to PFLOTRAN Initialize, execute and finalize. Surely, CLM should only link to PLFOTRAN initialize, when then links to PFLOTRAN execute, then finalize. Also, according to your diagram, no information flows back from PFLOTRAN to CLM? Your diagram suggests that there is no two-way coupling? But the text state that soil moisture and hydraulic properties from PFLOTRAN and given back to CLM. Your flowchart does not really show this? Use m day-1 rather than m d-1.

**Response:** In response to all reviewers' comment regarding model coupling, we have significantly revised the technical details about model coupling in Section 2.3 (see below as well) and added a new schematic describing the model coupling (Figure 1) in the revised manuscript. We have updated Figure 1 to better represent the two-way model coupling. We also have changed the unit "m d-1" to "m day-1".

While the updated section addresses all of the questions raised by the reviewer, we summarize the answers here. The model coupling interface is able to accommodate different vertical and horizontal resolution between the two models. In our present work, both models had the same horizontal resolution, but different vertical resolution. CLM used the default, exponentially varying vertical discretization, but PFLOTRAN had a uniform 0.5 [m] vertical spacing. The vertical extent of the domain in PFLOTRAN is deeper than the CLM domain. The model coupling interface uses conservative interpolation scheme to remap data between two model grids.

Figure 4 – Sorry I can hardly read any of the figure titles, please make these larger and more easily readable.

**Response**: Thanks for pointing this out. We have made the font of the figure titles bigger in the revised manuscript, including those in supplementary materials (figures S1 and S2).

Lines 359 – 361: You state that cold month were excluded from the analysis as you end up with division by zero issues when LH becomes close to zero. That's why most people use the evaporative fraction (EF), rather than the Bowen ratio. With EF, you take the ratio of latent to the sum of sensible and latent, hence, you will not have division by zero issues. You should use EF rather than Bowen ratio.

**Response**: Thanks for the great suggestion. We have redone the analysis using EF instead of Bowen ratio, and modified the figure and text in the revised manuscript correspondingly.

Section 4.1 – Please use model evaluation rather than model validation. Validation implies the model is already correct to start with and you are therefore validating it. This is of course never true of any model.

**Response:** Thanks for the great suggestion. We have changed the section title to be "Model evaluation".

Line 418: Looking at Figure 7(a) and 8, I find it hard to get an accurate idea of the differences, could you please plot the difference instead? Figure 10 – can you please remove the textbox at the bottom (CONTOUR FROM......). Looks like an NCL plot to me I'm sure you can remove this: ares@cnInfoLabelOn = False

**Response**: We have modified the figures as suggested. Please check the revised manuscript for details. We also modified the figures in the supplementary material accordingly to be consistent.

Line 447: Don't start a sentence with And. Your use of the 2 m simulation as a surrogate truth is fine, given a lack of observations of what is being simulated. However, you cannot really say simulation x outperformed simulation y (line 482), explain why one simulation appears more realistic, but I am not comfortable with the word "outperform". It would have been really interesting if you ran your model over a site for which observationally derived flux tower estimates of H and LE are available, such that you could then assess if this coupled system actually improves on CLM4.5 alone in simulating surface energy fluxes. I do understand that locations where Flux tower data are currently available (e.g., the FLUXNET network), may not necessarily be regions where the hydrology is interesting enough to warrant the use of such a model. You do however, need to acknowledge somewhere that the model needs to be evaluated against actual observations of surface fluxes.

**Response:** Thanks for the great suggestions. We have\_modified the sentences as suggested in the revised manuscript. In fact, two flux towers have been installed along the Hanford reach for this purpose but the analysis of the flux measurement is still preliminary. In addition, both towers are a little distant from the modeling domain to satisfy the requirements of eddy covariance measurements. Nevertheless, we also added discussions on the need of evaluating the model using eddy covariance measurements in lines 622-627 of the revised manuscript.

Code availability: We had a recent discussion among GMD editors, and the point of the Code Availability section is to ensure the reproducibility. What we want is the exact code used for this paper. It is of course understandable that the code is still under development, however, we request you make the version of the code used for this paper available. If this is already on bitbucket or github, it is quite easy to make the revision/branch used for this study on ZENODO, which is the preferred repository for code as per GMD guidelines as it will generate an actual DOI for the code: http://www.geoscientific-modeldevelopment.net/about/code\_and\_data\_policy.html

If you do a quick search on ZENODO, you will find several codes which point to github/bitbucket repositories, but a "frozen" version of the code used can be directly obtained from ZENODO, rather than a user having to work out which branch/revision of your code was used in the paper from the github/bitbucket repo

**Response**: Thanks for the instruction. The model and data have been made publicly available at

- https://bitbucket.org/clm\_pflotran/clm-pflotran-trunk: CLM code
- https://bitbucket.org/clm\_pflotran/pflotran-clm-trunk: PFLOTRAN code
- https://bitbucket.org/pnnl\_sbr\_sfa/notes-for-gmd-2017-35: Data

The README file in the notes-for-gmd-2017-35 repository provides detailed notes on how to create, compile, and run a simulation. This information has been incorporated into the "code availability" section of the revised manuscript. Once the manuscript is accepted, we will start porting the frozen version of the code to ZENODO.

1Lawrence Berkeley National Laboratory, Berkeley, CA

2Pacific Northwest National Laboratory, Richland, WA

[revised manuscript text omitted]

$$\boldsymbol{q} = -\frac{kk_r}{\mu} \nabla(\boldsymbol{p} - \rho g \boldsymbol{z}), \tag{2}$$

with waterliquid pressure p, viscosity $\mu$ , acceleration of gravity g, intrinsic permeability k, relative permeability  $k_r$  and elevation above a given datum z. Conservative solute transport in the liquid phase is based on the advection-dispersion equation

$$\frac{\partial}{\partial t}(\varphi sC) + \nabla \cdot (\boldsymbol{q} - \varphi sD\nabla)C = 0, \qquad (3)$$

with solute concentration *C* and hydrodynamic dispersion coefficient *D*. PFLOTRAN employs backward Euler time discretization and finite volume spatial discretization. The discrete

systemdiscretized set of nonlinear equationsPDEs for flow and transport are solved by using the Newton-Raphson method.

**2.3 Model coupling**

For the coupled PFLOTRAN\_CLM v1.0 modelIn this workstudy, CLM4.5's one-dimensional models for flow in unsaturated [Zeng and Decker, 2009] and saturated [Niu et al., 2007] zones are replaced by PFLOTRAN's RICHRADS mode to simulate unsaturated-saturated flow within the describes the subsurface flow and solute transportthree-dimensional subsurface domain. while CLM4.5 simulates the land surface processes. AltThough7 PFLOTRAN is also capable of simulating coupled flow and thermal processes in the subsurface including explicit representation of liquid-ice phase [Karra et al., 2014], as well as, The-soil carbon and nitrogennutrient cyclesing, [Hammond and Lichtner, 2010; Zachara et al., 2016; Tang et al., 2016], can be solved in either PFLOTRAN or CLM4.5 but those processes are not enabled coupled between the two models in this study. A schematic representation of the coupling between PFLOTRAN and CLM4.5 (within the CESM1 framework) is shown in top panel of Figure 1.

A schematic representation of the coupling between CLM4.5 and PFLOTRAN is shown in Figure 1. -A model coupling interface layer-based on PETSc data structures was was developed to couple PFLOTRAN and CLM4.5, the two models and the interface includinges some key design features of the CESM coupler [*Craig et al.*, 2012]. The model coupling interface to supportallows each model grid to have a different spatial resolution and domain decomposition across multiple processors. (i) different model domain decompositions and (ii) different grid resolutions. Both models support distributed memory parallelism via MPI, but perform domain decomposition across multiple processors differently. While CLM4.5 uses a round-robin decomposition approach, PFLOTRAN employs domain decomposition via PETSc (Figure 1a). Interpolation of gridded data from one model's grid onto the grids of the otherother's grid is done through sparse matrix vector multiplication. 
[revised manuscript text omitted]
  $\frac{PFCLM_{E2m}}{S_{E2m}}$  (figures Figure 8-9 and 910) was significantly shifted from that in PFCLM2m-S2m (Figure 7a8a) through two mechanisms: (1) expanding the periodically inundated fraction of the riparian zone (i.e., surface elevation  $\leq 110$  m); and (2) enhancing moisture availability in the vadose zone in the inland domain (i.e., surface elevation > 110 m) through capillary rise. Both mechanisms led to general increases in simulated vadose-zone moisture availability and therefore higher latent heat fluxes compared to the simulations driven by the observed condition. For the inland domain, Bowen ratios in the warm seasonevaporative fraction clearly displayed a decliningan increasing trend as the groundwater table level increased (i.e., shallower)becomes shallower, consistent between the simulations (Figure 9a10c). 75% of tThe daily Bowen ratioevaporative fractionss for the inland domain stayed  $\frac{\text{mostly} > 5.0 \text{ well below } 0.2}{\text{when the water table levels are less than}}$ 108-112 m, suggesting decoupled surface-subsurface conditions in a typical semi-arid environment. When water table levels increased to be above 108-112 m, the coupling between the land surface energy budget and groundwater dynamics became strongerthe evaporative fraction increases to ~0.2. As, the elevation of the land surface is around 114-115 m, indicating that the water table fluctuated within the 6 m to 7 m range from the land surface, the surface and subsurface processes were become more strongly coupled, consistent with literature findings [Leung et al., 2011; Maxwell and Kollet, 2008]. Consequently, 50-75% of the daily Bowen ratio values stayed well below 5.0 because of improved water availability for evapotranspiration, especially in the elevated simulation (i.e., PFCLME2mSE2m). Bowen-Evaporative fraction ratios-in the riparian zone remained within the range of [-1.0, 1.0]close to 1.0, suggesting strong influences of the river and the role of deeper rooted plant types (e.g., riparian trees and shrubs) in modulating the energy partitioning (Figure 910d) of riparian zones in the semi-arid to arid environments.

To confirm the above findings, the liquid saturation [*unitless*] and mass of river water [*mol*] in the domain from  $PFCLM_{2m}-S_{2m}$  and  $PFCLM_{E2m}-S_{E2m}$  on 30 June each year are plotted along a transect perpendicular to the river (y = 200 m) in figures 10-11 and S4S7, and across a x-y plane

at an elevation of 107 m in figures S5-S8 and S6S9, respectively. Driven by the pressure introduced by elevated river stages, river water not only intruded further toward or even across the western boundary in high water years, but also led to shallower water table and increased liquid saturation in the vadose zone due to capillary rise across the domain. In fact, liquid saturation in the shallow vadose zone could increase from 0.1-0.2 in PFCLM2m-S2m to 0.3-0.4 in PFCLME2m-SE2m on these days because of river water intrusion. And tThe river-water tracer could show up in the near-surface vadose zone at a distance of ~400 m from the river (Figure S4S7). Interestingly, by comparing the spatial distributions of river-water tracer in the low-water year (i.e., 2015) between the "observed" and "elevated" scenarios, the presence of river water in the domain was much less in the elevated scenario in terms of its spatial coverage (figures 10-11and S4S7). This pattern suggests that after a number of years of enhanced river water intrusion into the domain, the hydraulic gradient between groundwater and river-water could be reversed, so that groundwater discharging might be expected more frequently in low-water years in a prolonged elevated scenario.

The responses of LH and Bowen ratioevaporative fraction (figures 8–9 and 910) indicated that a tight coupling among stream, aquifer, and land surface processes occurred in the elevated scenario, which could become realistic in one to two decades for the study site, or for other sites along the Hanford reach characterized by lower elevations under the current condition.

**4.3 Effect of spatial resolution**

To apply the model to large-scale simulations or over a long time period, it is important to assess how the model performs at coarser resolution, as the 2-m simulations are computationally expensive. Here, we use the 2-m simulations (i.e.,  $PFCLM_{2m}-S_{2m}$  and  $PFCLM_{E2m}S_{E2m}$ ) simulations as benchmarks for this assessment. That is,  $PFCLM_{2m}-S_{2m}$  and  $PFCLM_{E2m}-S_{E2m}$ simulated variables are treated as the "truth" for "observed" and "elevated" river stage scenarios, and outputs from other simulations are compared to them to verify their performance. In the previous section, we showed that simulated water table levels from the model were virtually identical to observations. In this section, we further quantify biases of other variables of interest from the high-fidelity 2-m simulations.

The domain-averaged daily surface energy fluxes from PFCLM2m-S2m show clear seasonal patterns, which are consistent in terms of their magnitudes and timing, reflecting mean climate conditions at the site (Figure S6S10). Driven by elevated river stages, latent heat from **PFCLME2m-SE2m** is consistently higher than that from **PFCLM2mS2m**. The mean latent heat and sensible heat fluxes simulated by  $\frac{PFCLM_{2m}-S_{2m}}{W}$  were 14.1 W m-2 and 38.7 W m-2 over this period, compared to by 18.50 W m-2 and 35.75 W m-2 in PFCLME2mSE2m. Figure 11-12 shows deviations of simulated LH and SH in the 20-m and 10-m simulations from the corresponding 2m simulations. The deviations of both LH and SH were small across all the simulations driven by the observed river stage when surface and subsurface were decoupled. In the elevated simulations (i.e., PFCLME10m-SE10m and PFCLME20mSE20m) when surface and subsurface processes are more tightly coupled, errors in surface fluxes became significant in the coarse resolution simulations when compared to  $\frac{PFCLM_{E2m}S_{E2m}}{S_{E2m}}$ . For example, the relative errors in LH were 2.41% and 1.35% for PFCLM20m-S20m and PFCLM10mS10m, respectively, as compared to PFCLM2mS2m, but grew as large as 33.84% and 33.19% for PFCLMF20m-SE20m and **PFCLME10mSE10m**, respectively, when compared to **PFCLME2mSE2m**. The 10-m simulations outperformed the 20-m simulations under both scenarios but the magnitudes of errors were comparable. On the other hand, notably the vertical only simulation ( $\frac{PFCLM_{v2m}S_{v2m}}{S_{v2m}}$ ) has a small error of 5.67% in LH compared to PFCLM2mS2m, indicating that lateral flow is less important when water table is deep.

To better understand how water in the river and the aquifer was connected, we also quantified the biases of subsurface state variables and fluxes including total water mass and tracer amount, as well as exchange rates of water and tracer at four boundaries of the subsurface domain using a similar approach (Figure  $\frac{S7-S11}{1}$  and Figure  $\frac{1213}{2}$ ). Compared to the magnitude of total water mass in the domain (averaged 919.45 ×106 Kg and 1020.19 ×106 Kg in PFCLM2m  $\underline{S}_{2m}$  and PFCLME2mSE2m), errors introduced by coarsening the resolution were very small under the observed river stage condition (0.04% for PFCLM20m-S20m and 0.03% for PFCLM40mS10m) and grew to 9.85% for PFCLME20m-SE20m and 9.87% for PFCLME10m-SE10m in terms of total water mass in the domain (Table 5). However, for total tracer in the domain (averaged 142.07×106 mol and 172.46 ×106 mol in PFCLM2m-S2m and PFCLME2mSE2m) as a result of transport of river water in lateral and normal directions to the river, resolution clearly makes a difference under both observed condition and elevated scenarios (relative errors of 5.44% for

**PFCLM**10m**S**10m, 10.40% for **PFCLM**20m**S**20m, and 22.0% for both **PFCLM**E10m-**S**E10m and **PFCLM**E20m**
[revised manuscript text omitted]

| PFCLM 2m S 2m     | CP v1.0 | 2m                       | Yes          | Observed        |
| PFCLM 10m S 10m   | CP v1.0 | 10m                      | Yes          | Observed        |
| PFCLM 20m S 20m   | CP v1.0 | 20m                      | Yes          | Observed        |
| PFCLM E2m S E2m   | CP v1.0 | 2m                       | Yes          | Observed +5     |
| PFCLM E10m S E10m | CP v1.0 | 10m                      | Yes          | Observed +5     |
| PFCLM E20m S E20m | CP v1.0 | 20m                      | Yes          | Observed +5     |
| CLM2m                 | CLM4.5  | 2m                | No           | Not applicable  |

Table 1. Summary of numerical experiments

| Material | Porosity | Permeability              | Van Genuchten/Burdine Parameters |      |                       |  |
|----------|----------|---------------------------|----------------------------------|------|-----------------------|--|
|          |          | ( m 2 ) | Res. Sat.                        | m    | alpha                 |  |
| Hanford  | 0.20     | 7.387×10 -9    | 0.16                             | 0.34 | $7.27 \times 10^{-4}$ |  |
| Ringold  | 0.40     | 1.055×10 -12   | 0.13                             | 0.75 | 1.43×10 -4 |  |

Table 2. Hydrogeological material properties of Hanford and Ringold materials.

| Well      | PFCLM 2m S 2m |       | PFCLM 10m | 510m | PFCLM 20m S 20m |       |  |
|-----------|-------------------------------------|-------|----------------------|-------------|---------------------------------------|-------|--|
| number    | RMSE (m)                            | N-S   | RMSE (m)             | N-S         | RMSE (m)                              | N-S   |  |
| 399-3-29  | 0.022                               | 0.999 | 0.022                | 0.999       | 0.021                                 | 0.999 |  |
| 399-3-34  | 0.011                               | 1.000 | 0.011                | 1.000       | 0.006                                 | 1.000 |  |
| 399-2-01  | 0.039                               | 0.997 | 0.038                | 0.997       | 0.029                                 | 0.998 |  |
| 399-1-60  | 0.016                               | 1.000 | 0.016                | 0.999       | 0.013                                 | 1.000 |  |
| 399-2-33  | 0.028                               | 0.998 | 0.028                | 0.998       | 0.022                                 | 0.999 |  |
| 399-1-21A | 0.023                               | 0.999 | 0.023                | 0.999       | 0.020                                 | 0.999 |  |
| 399-2-03  | 0.037                               | 0.997 | 0.037                | 0.997       | 0.029                                 | 0.998 |  |
| 399-2-02  | 0.045                               | 0.995 | 0.045                | 0.995       | 0.042                                 | 0.996 |  |
| mean      | 0.028                               | 0.998 | 0.028                | 0.998       | 0.023                                 | 0.999 |  |

| Table 3. | The comparison | between | simulated | and | observed | water | table | levels |
|----------|----------------|---------|-----------|-----|----------|-------|-------|--------|
|          |                |         |           |     |          |       |       |        |

Table 4. The relative error in surface energy fluxes simulated by  $\frac{PFCLM_{10m}-S_{10m}}{S_{10m}}$  and  $\frac{PFCLM_{20m}-S_{20m}}{S_{20m}}$  benchmarked against  $\frac{PFCLM_{2m}-S_{2m}}{S_{2m}}$  and by  $\frac{PFCLM_{E10m}-S_{E10m}}{S_{E10m}}$  and  $\frac{PFCLM_{E20m}-S_{E20m}}{S_{E20m}}$  benchmarked against  $\frac{PFCLM_{E2m}-S_{2m}}{S_{E2m}}$

| Simulation                              | Latent heat flux (%) | Sensible heat flux (%) |
|-----------------------------------------|----------------------|------------------------|
| PFCLM v2m S v2m   | 5.67                 | 1.63                   |
| PFCLM 10m S 10m   | 1.35                 | 0.78                   |
| PFCLM 20m S 20m   | 2.41                 | 1.42                   |
| PFCLM E10m S E10m | 33.19                | 13.71                  |
| PFCLM E20m S E20m | 33.84                | 14.18                  |

Table 5. The relative error in total water mass and tracer amount in the subsurface simulated  $\frac{by-in}{PFCLM_{10m}-S_{10m}}$  and  $\frac{PFCLM_{20m}-S_{20m}}{PFCLM_{E10m}-S_{20m}}$  benchmarked against  $\frac{PFCLM_{2m}-S_{2m}}{PFCLM_{E20m}-S_{E20m}}$  benchmarked against  $\frac{PFCLM_{E2m}-S_{2m}}{PFCLM_{E2m}-S_{E2m}}$

| Simulation                              | Total water mass (%) | Total tracer (%) |
|-----------------------------------------|----------------------|------------------|
| PFCLM 10m S 10m   | 0.03                 | 5.44             |
| PFCLM 20m S 20m   | 0.04                 | 10.40            |
| PFCLM E10m S E10m | 9.87                 | 22.00            |
| PFCLM E20m S E20m | 9.85                 | 22.00            |

---

## Author Response (AR2)

**Coupling a three-dimensional subsurface flow and transport model with a land surface model to simulate stream-aquifer-land interactions (CPv1.0) [MS No.: gmd-2017-35]**

**Responses to review comments**

Topical Editor Decision: Reconsider after major revisions (21 Sep 2017) by Jatin Kala Comments to the Author:
Dear authors,

Apologies for the delay, a second review never came back. I have reviewed your responses, but i remain unconvinced about the reasons you outline for not comparing CPv1.0 fluxes with CLM stand alone fluxes. These are of course different models, making different assumptions. But this is not a valid reason for not explaining the differences. You simply refer the reader to Figure S4, and leave the reader to make their own conclusions. The differences in latent heat flux between CPv1.0 and CLM stand alone are very large. You need to explain why these differences are so large. And which is likely to be more "correct/realistic"???

If you were to use CLM versus CPv1.0 coupled to an atmospheric model, such large differences in latent heat flux would have a very large influence on near surface temperature, humidity, boundary-layer structure etc. I do not think you can leave this analysis out. You have added/coupled an additional component to CLM, it makes sense to me, that you should compare simulations with the additional component, to the original model. That would generally be expected for any model development. A reader of this manuscript will be left very confused as to why the latent heat fluxes are so different between the two. You have to address this explicitly in the manuscript.
Kind regards,

Jatin
**Response**:

Dear Editor,

Thank you very much for your suggestions. We greatly appreciate your effort. We agree with your comments that it is important to show the difference between CP v1.0 and CLM to illustrate how model assumptions would affect the simulations, and provide explanations to mechanisms leading to such differences.

In response to your comments, we made the following changes in the revised manuscript:

- Two new figures are added:
    - Figure 12 that compares spatially-averaged key hydrologic fluxes and states simulated by CLM and CPv1.0 for the study period.
    - Figure S9 that shows spatial maps of CLM4.5 simulated latent heat flux for the month of June during the study period, and their differences from that of $S_{2m}$.
- Differences in simulation results of CLM and CP v1.0 are discussed in the revised manuscript in sections 4.2 and 5 on pages 18, 19, and 22.

- Additionally, corresponding changes are made to the abstract (page 2) and section 3.2 (page 15).

We sincerely hope that our revisions are satisfactory so that the manuscript can be accepted for publication. Thanks again and we look forward to hearing from you.

Maoyi on behalf of the co-authors

**Coupling a three-dimensional subsurface flow and transport model with a land surface model to simulate stream-aquifer-land interactions (CP v1.0)**

Gautam Bisht[1], Maoyi Huang[2,*], Tian Zhou[2], Xingyuan Chen[2], Heng Dai[2], Glenn E. Hammond[3], William J. Riley[1], Janelle L. Downs[2], Ying Liu[2], John M. Zachara[2]

[1]Lawrence Berkeley National Laboratory, Berkeley, CA

[2]Pacific Northwest National Laboratory, Richland, WA

[3]Sandia National Laboratories, Albuquerque, NM

Correspondence to: Maoyi Huang (maoyi.huang@pnnl.gov)

Revised Manuscript to be considered for *Geoscientific Model Development*

**Abstract**

A fully coupled three-dimensional surface and subsurface land model is developed and applied to a site along the Columbia River to simulate three-way interactions among river water, groundwater, and land surface processes. The model features the coupling of the Community

Land Model version 4.5 (CLM4.5) and a massively-parallel multi-physics reactive transport model (PFLOTRAN). The coupled model, named CP v1.0, is applied to a 400 m × 400 m study domain instrumented with groundwater monitoring wells along the Columbia River shoreline.

CP v1.0 simulations are performed at three spatial resolutions (i.e., 2 m, 10 m, and 20 m) over a five-year period to evaluate the impact of hydro-climatic conditions and spatial resolution on simulated variables. Results show that the coupled model is capable of simulating groundwater- river water interactions driven by river stage variability along managed river reaches, which are of global significance as a result of over 30,000 dams constructed worldwide during the past half century.  Our numerical experiments suggest that the land-surface energy partitioning is strongly modulated by groundwater-river water interactions through expanding the periodically inundated fraction of the riparian zone, and enhancing moisture availability in the vadose zone via capillary rise in response to the river stage change. Meanwhile, CLM4.5 fails to capture the key hydrologic process (i.e., groundwater-river water exchange) at the site, and consequently simulates drastically different water and energy budgets. 
[revised manuscript text omitted]

As discussed in sections 2.1 and 3.2., the hydrologic parameterizations in the default CLM4.5

model are based on conceptual and physical understandings from watershed hydrology that do not apply at the scale of our study site, where the exchange of river water and groundwater dominates the hydrologic budget of the system. Nevertheless, a comparison between CLM4.5

and CP v1.0 helps characterize how scale inconsistencies in physical representations affect the simulations. Figure 12 shows comparisons of key components in the hydrologic budget between the two models. The simulated mean water table elevation of the domain from CLM4.5  ranges between 74 m and 80 m (i.e., 35 m – 40 m below the surface), while the observed water table elevation ranges between 104 m and 108 m (i.e., 5 m – -10 m below the surface), and was accurately reproduced by $S_{2m}$ (Figure 12a). By using physics derived for the larger scale,

CLM4.5 could not capture subsurface river water and groundwater exchanges, and consequently cannot accurately simulate groundwater table dynamics for our study domain.

At this semi-arid field site, the groundwater and  river water exchanges represented in $S_{2m}$ recharges the unconfined aquifer, and hence maintains sufficient soil water availability in the top 3.8 m of the soil column, while the lack of groundwater and river water interactions in $CLM_{2m}$ leads to overall declining soil water content with seasonal variability as a result of percolation of winter rain water (Figure 12b). The difference in soil moisture availability propagates to evapotranspiration (ET) and its components (figures 12c-f). Simulated summer ET in $CLM_{2m}$ shows a high-frequency signal in response to rainfall pulses through ground evaporation. Transpiration simulated by $CLM_{2m}$ is determined by soil water availability in the soil column. In the spring and early summer of 2011 and 2013, transpiration from $CLM_{2m}$ is close to that from $S_{2m}$ given sufficient soil water. For other periods, $CLM_{2m}$ simulates significant lower transpiration rates compared to $S_{2m}$.

Simulated latent heat fluxes in June for the period of 2011-2015 from $CLM_{2m}$ and their differences from those in $S_{2m}$ are also illustrated in figures S9a and b. Evidently, the hydrologic gradient from river to inland is missing as CLM4.5 lacks the capability of capturing the river stage dynamics at such a resolution (in Figure S9a). Instead, even though initiated from the same initial condition as $S_{2m}$ on 01/01/2009 as discussed in the spin-up procedure in section 3.2, soil moisture at the grid cells inundated or periodically inundated by the river is soon depleted through ET, surface runoff, or baseflow. On the other hand, latent heat from the inland domain is generally higher in $CLM_{2m}$ than in $S_{2m}$ due to ground evaporation in response to rainfall pulses. In short, CLM4.5 fails to capture the dynamics of groundwater and river water exchanges. These biases propagates to simulated water and energy fluxes, which could have large impacts on boundary layer evolution, convection, and cloud formation in coupled land-atmosphere studies.

[revised manuscript text omitted]
 comparing simulations from the coupled model (CPv1.0) to that from CLM4.5, we demonstrated that the catchment-scale physics imbedded in CLM4.5 does not apply at the field scale. By misrepresenting, or not including, key hydrologic processes at the scale of interest, CLM4.5 fails to capture groundwater table dynamics, which could propagate to water and energy budgets and have profound impacts on boundary layer, convection, and cloud formation in coupled land-atmosphere studies. Our finding is consistent with results from other recent studies in which integrated surface and subsurface models were compared to standalone land surface models [*Fang et al.*, 2017; *Niu et al.,* 2017].

[revised manuscript text omitted]

DOE under contract DE-AC05-76RLO1830. We greatly appreciate the constructive comments from two anonymous reviewers and the topical editor, Dr., Jatin Kala, which helped improve the manuscript significantly.

[revised manuscript text omitted]